# Investigating the neuronal role of the proteasomal ATPase subunit gene *PSMC5* in neurodevelopmental proteasomopathies

Neurodevelopmental proteasomopathies are a group of disorders caused by variants in proteasome subunit genes, that disrupt protein homeostasis and brain development through poorly characterized mechanisms. Here, we report 26 distinct variants in *PSMC5*, encoding the AAA+ ATPase subunit PSMC5/RPT6, in individuals with syndromic neurodevelopmental conditions. Combining genetic, multi-omics and biochemical approaches across cellular models and *Drosophila*, we unveil the essential role of proteasomes in sustaining key cellular processes. Loss of PSMC5/RPT6 function impairs proteasome activity, leading to protein aggregation, disruption of mitochondrial homeostasis, and dysregulation of lipid metabolism and immune signaling. It also compromises synaptic balance, neuritogenesis, and neural progenitor cell stemness, causing deficits in higher-order functions, including learning and locomotion. Pharmacological targeting of integrated stress response kinases reveals a mechanistic link between proteotoxic stress and spontaneous type I interferon activation. These findings expand our understanding of proteasome-dependent quality control in neurodevelopment and suggest potential therapeutic strategies for neurodevelopmental proteasomopathies.

The proper structure and function of eukaryotic cells rely on the maintenance of the intracellular proteome composed of thousands of proteins. The dynamic balance between protein synthesis and degradation is orchestrated by the proteostatic network[1,2], an integral component of which is the ubiquitin-proteasome system (UPS), that selectively eliminates short- and long-lived proteins, as well as misfolded proteins modified with ubiquitin[3]. At the very heart of the UPS is the 26S proteasome, a large multi-subunit protease consisting of a 20S core particle and a 19S regulatory particle, comprising a lid and a base[3]. The lid subunits recognize and bind polyubiquitinated target proteins[1], while the base subunits Rpt1-6/PSMC1-6 utilize ATP hydrolysis to deubiquitinate, unfold and translocate substrates into the proteolytic chamber of the 20S core particle, where they are degraded into oligopeptides. These oligopeptides can be further hydrolyzed into amino acids by peptidases, enabling efficient amino acid recycling within the cellular protein synthesis machinery[1,3].

The remarkable dynamics and plasticity of proteasomes impart a significant impact on cellular physiology[3], particularly in neurons. Neuronal synaptic plasticity relies on continuous proteome renewal, necessitating robust translation at ribosomes and concurrent breakdown of defective ribosomal products[2,4]. This maintenance of protein homeostasis is particularly challenging in cortical neurons, where synapses may contain hundreds of proteins[4]. Effective protein clearance by the proteasome is crucial for various neuronal processes, including synaptic remodeling, cell migration, neurotransmitter release, long-term potentiation, long-term depression, and memory formation[5,6]. Inhibition of proteasome activity has been experimentally shown to disrupt synapse composition, promote the aggregation of polyubiquitinated proteins, and ultimately lead to neurodegeneration[6]. Similar observations have been made in brain tissues from individuals with neurodegenerative disorders[6] and schizophrenia[7]. Genetic investigations have further unveiled that loss-

✉e-mail: sebastien.kury@chu-nantes.fr; stephane.bezieau@chu-nantes.fr; frederic.ebstein@univ-nantes.fr; elke.krueger@uni-greifswald.de

of-function variants in proteasome genes contribute to the onset and progression of a unique class of neurodevelopmental disorders (NDDs), specifically referred to as the neurodevelopmental proteasomopathies (NDPs)[8] or proteasome associated neurodevelopmental disorders. Unlike proteasome associated autoinflammatory syndromes (PRAAS), which are typically caused by recessive 20S proteasome gene variants and characterized by severe autoinflammatory symptoms, NDPs primarily affect the central nervous system (CNS). The inheritance pattern for these disorders can either be dominant, for the majority of variants in *PSMD12* [Stankiewicz-Isidor syndrome, MIM: 617516][9,10], *PSMC3*[11] and *PSMD11*[12], or recessive, as observed in cases involving *PSMB1* [MIM: 620038][13] and *PSMC3* [MIM: 619354][14] variants (Supplementary Fig. 1).

In this context, the proteasome subunit PSMC5/Rpt6 is of particular interest due to its role in regulating proteasome activity and synaptic remodeling. As an ATPase, PSMC5 serves as a neuronal sensor that dynamically adjusts proteasome activity and localization to spines in response to protein turnover requirements through phosphorylation by CAMKIIα[15,16]. The present study further confirms the pivotal role of *PSMC5* in neuronal function. We describe 26 variants in *PSMC5*, the vast majority heterozygous and de novo, in 44 subjects, including 42 unrelated propositi with delayed neurodevelopment and two affected relatives. Functional analyses performed on patient cells and animal models unveiled that *PSMC5* alterations lead to a loss of proteasome function, triggering a complex cellular program involving profound remodeling of innate immunity, inflammation, and lipid metabolism. Intriguingly, the integrated stress response (ISR) mediates most of these changes, and potential molecular targets have been identified, holding promise for the development of biomarkers and therapeutic interventions. This study offers critical insights into the multifaceted impact of *PSMC5* variants on NDD, advancing our understanding of the intricate cellular mechanisms underlying these conditions.

## Results

### Rare pathogenic *PSMC5* variants cause a syndromic neurodevelopmental disorder

We identified 44 individuals with similar clinical features who carried a rare single nucleotide variant (SNV) or indel in *PSMC5*. Thirty-three cases were de novo, six heterozygous of uncertain inheritance, two inherited from an affected mother also enrolled, and one with biallelic variation. In total, we identified 26 distinct variants, most of them missense (19/26) impacting regions highly conserved from human to yeast, and four of them recurring (Fig. 1a and Supplementary Fig. 2). They were predominantly located in the "ATPases Associated with diverse cellular Activities (AAA-ATPase)" domain (16/26); the recurring variants NM_002805.6: c.959C>G p.(Pro320Arg) (6/44 subjects) and c.973C>T p.(Arg325Trp) (9/44) were located near the end of the domain (Fig. 1a).

Our three-dimensional structural analysis showed notably that about half of the variants primarily located in the lower ATPase domain interface with neighboring base ATPase subunits PSMC1/Rpt2 and PSMC4/Rpt3, and the substrate in its engaged state (Fig. 1 and Supplementary Fig. S3). They are likely to affect proteasome function in multiple ways, by disrupting proteasomal complex formation, impairing the transition from substrate-free to substrate-engaged states of the 26S proteasome, affecting ATP binding and hydrolysis, hindering substrate processing, or perturbing protein unfolding and subunit incorporation (Fig. 1; Supplementary Figs. 3 and 4). Overall, these findings, detailed in supplemental notes, concur with the simulated classification of the identified *PSMC5* variants according to the American College of Medical Genetics and Genomics/Association for Molecular Pathology (ACMG/AMP) guidelines, suggesting pathogenicity for 25/26 variants (Supplementary Table 1).

Abnormal neurodevelopment was observed in all subjects, with global developmental delay present in 42/44 (95%) cases (Fig. 2a; Supplementary Table 2). Neurological manifestations included absence or delay in speech (36/39; 92%), intellectual disability (26/32; 81%), motor delay or impairment (32/41; 78%), abnormal behavior (30/39;77%, including autism spectrum disorder (ASD) and attention deficit hyperactivity disorder (ADHD)), abnormal muscle tone (27/38; 71%), and seizures (14/43; 33%) (Fig. 2a; Supplementary Table 2). Brain magnetic resonance imaging revealed frequent abnormalities (14/29; 48%) ranging from non specific myelination delay to brain atrophy and malformations, such as corpus callosum, pons and cerebellar hypodysplasia. The phenotype appeared syndromic (Fig. 2a; Supplementary Table 2), with notable non-neurological findings such as ophthalmological anomalies (27/37; 73%), skeletal malformations (26/42; 62%), feeding difficulties (23/41; 56%), cardiac abnormalities (13/41; 32%), hearing loss (11/39; 28%, conductive in five individuals, sensorineural and mixed in one patient each), genital abnormalities (7/41; 17%) and kidney abnormalities (5/37; 14%).

### Subjects with *PSMC5* variants present a similar facial gestalt

Most affected individuals exhibited abnormal facial shape (32/40; 80%), often accompanied by craniofacial abnormalities, including microcephaly (15/38; 39%) or abnormality of the mandible (14/39; 36%, including micrognathia, retrognathia and prognathia) (Fig. 2b). However, the manifestations were heterogeneous and did not allow conclusive diagnosis of the disorder through standard clinical assessment. Nonetheless, facial analysis with GestaltMatcher[17] suggested a recognizable facial gestalt among *PSMC5* subjects. A strong facial resemblance was indeed observed between the 15 *PSMC5* subjects tested, 90% of subject combinations having mean pairwise distances below the threshold ($c = 0.915$) (Supplementary Fig. 5). Analysis by pairwise comparison matrix extended to 7459 images from 449 disorders in GMDB revealed that 10 of 15 *PSMC5* subjects (S3/10/12/15/16/23/34-37) matched at least another *PSMC5* subject with a rank below 50 (Fig. 2c). Subject pairs/trios (S10, S35 and S37), (S12 and S35), and (S3 and S34) were highly similar, with ranks below ten, indicating strong resemblance.

### Variants affecting the AAA+ ATPase domain tend to induce a more severe phenotype

Even if a clear genotype-phenotype correlation is difficult to draw because of an important phenotypic variability across subjects, the multisystemic and syndromic characteristics of the disorder are particularly pronounced when variants are located within or in the immediate vicinity of the ATPase domain. This is especially true for the most recurrent variant in our cohort, c.973C>T p.(Arg325Trp), which is more likely to be associated with severe neurodevelopmental impact and heart defects, facial dysmorphic features, hearing loss, visual impairment and feeding difficulties. On the other hand, variants located outside the ATPase domain, affecting the first part of the protein, are frequently associated with short stature (Fig. 2c; Supplementary Table 2).

### *Psmc5* knockdown (KD) impairs fly reversal learning and alters proteostasis as well as excitatory/inhibitory balance of rat hippocampal neurons

Pan-neuronal ELAV-GAL4 KD of *Rpt6*, *Drosophila melanogaster*'s highly conserved orthologue of *PSMC5*, had no significant effect on classical olfactory learning, during which flies form association between an odor to foot shock (Tukey, $p = 0.3059$; $N = 4$ performance indexes (PI) per group; Fig. 3a). However, *Rpt6* KD caused significant defect in the ability to acquire novel association after initial training—a different behavior paradigm known as reversal learning (Tukey, $p < 0.0001$; $N = 4$; Fig. 3b), which is similar to what we previously reported in proteasome subunit

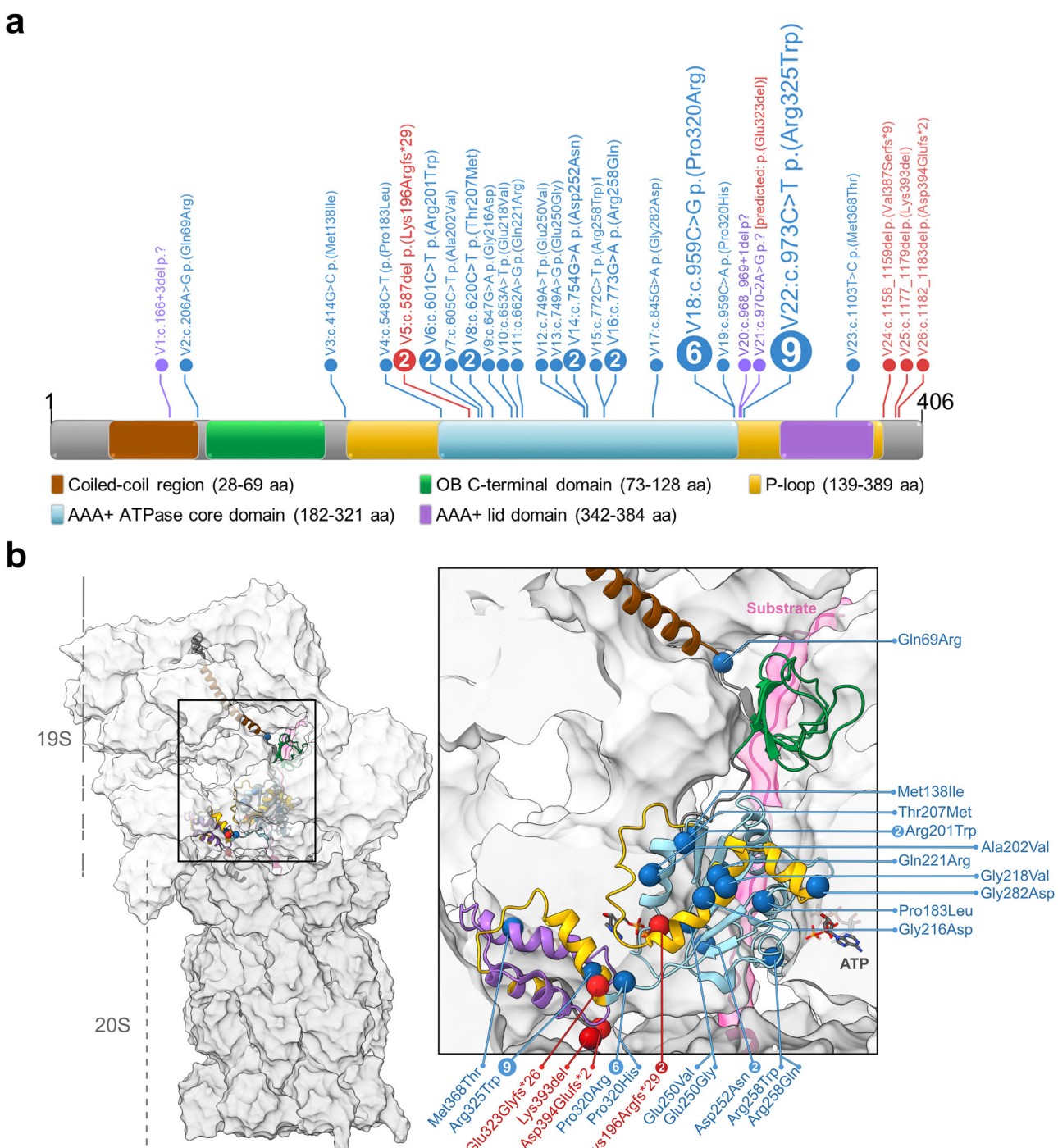

**Fig. 1 | Structural mapping of PSMC5 variants highlights clustering within the AAA⁺ ATPase domain. a** Schematic of the PSMC5/RPT6 protein showing 26 variants identified in NDD patients. Approximately half cluster within three hotspot regions of the AAA⁺ ATPase domain. Missense variants are shown in blue, frameshift/indels in red, and splice site variants in purple. Numbers indicate the count of unrelated individuals carrying each variant. **b** Structural localization of PSMC5 variants within the 26S proteasome (based on PDB: 6MSK). Variants (spheres) are primarily situated in the lower ATPase domain of RPT6. Portions of the complex were hidden to enhance visibility; RPT6 is shown in cartoon representation.

gene *PSMC3* KD models[11]. Interestingly, contrary to wild-type (WT) controls, performance after reversal learning did not change significantly at 1 week (6-8 days) and 3 weeks (22–24) (WT Tukey, $N = 4$ biological replicates) (Fig. 3c).

To determine if *Rpt6* KD led to uniform impact on behavior, we tested locomotion activity by leveraging the well-established climbing assay in *Drosophila*[18]. We observed a different profile for climbing than that for reversal learning in pan-neuronal *Rpt6* KD transgenics (Fig. 3d). While reversal learning deficiencies were similar from day 1 day up to

3 weeks, we observed that decline in climbing performance was initially stable for 1 week (Fig. 3e–h) but dropped markedly at 22 days (Fig. 3i, j). On the other hand, WT flies maintained normal performance during the same period.

We studied the effects of *Psmc5* loss on primary rat hippocampal neurons by gene KD, using *Psmc5*-specific shRNA or scrambled shRNA for controls (scrambled Ctrl) along with co-expressed GFP delivered via lenti-virus infection of cells at day in vitro 1 (DIV1). After 14 days in culture, a significant reduction of PSMC5 levels was measured by

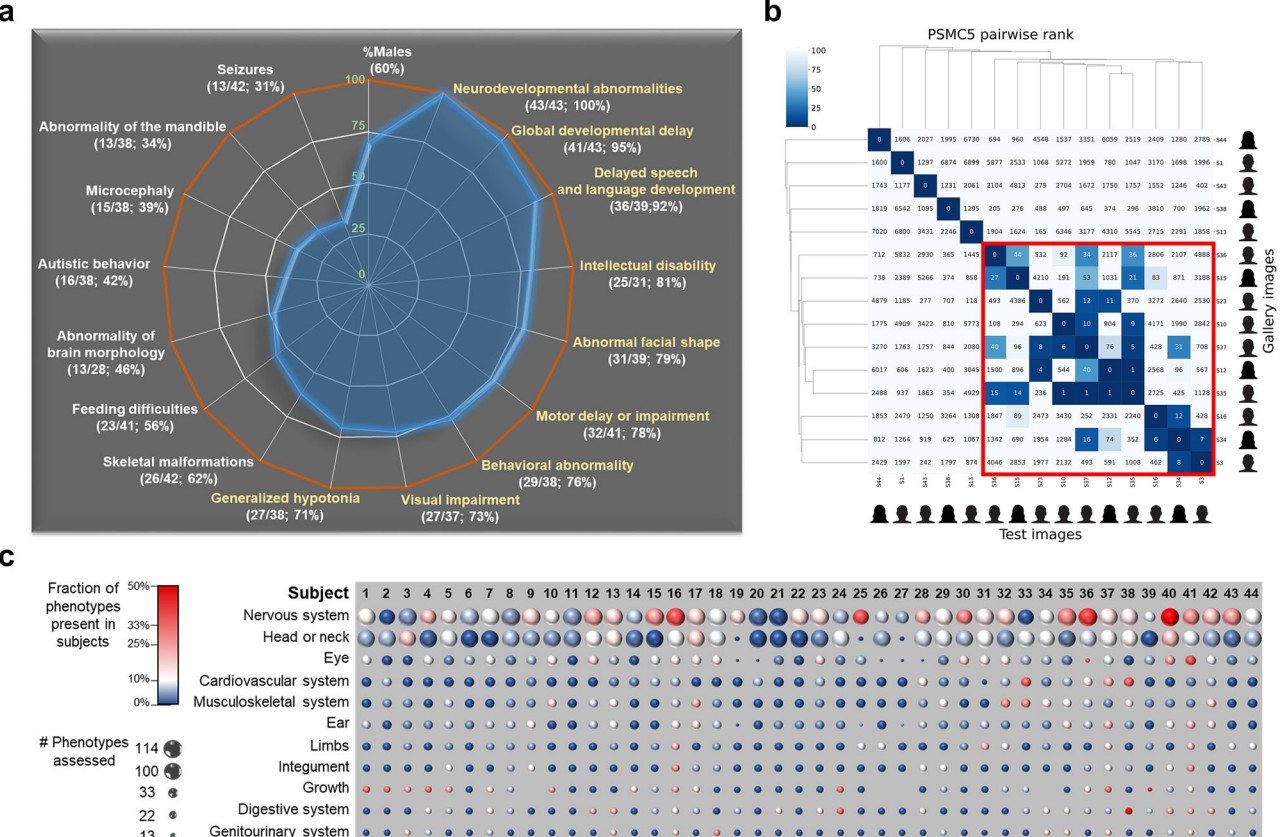

**Fig. 2 | Clinical features associated with *PSMC5* variants. a** Radar chart summarizing the most frequent clinical findings in the cohort. **b** Pairwise rank matrix and hierarchical clustering of 15 *PSMC5* subjects (avatars shown). A red box highlights individuals with at least one match ranked below 50, indicating shared facial features. **c** Visualization of HPO-based phenotypic categories across affected individuals. Circle size reflects the number of phenotypes assessed per subject; color indicates the proportion calculated (dark blue: low, red: high). Categories are ranked by average prevalence, with nervous system and head/neck anomalies most frequently observed.

fluorescent labeling with PSMC5-specific antibodies (Fig. 3k, l). Notably, under control conditions, PSMC5 labeling appeared predominantly in neurons and much less in glial cells.

*Psmc5* KD by shRNA significantly increased ubiquitin-positive protein levels throughout the cells, particularly in the soma. This accumulation of ubiquitin-conjugated proteins (Fig. 3m, n), reflecting impaired proteasomal activity, confirms that PSMC5 is essential for maintaining proper proteasomal function and regulating proteostasis in neurons. To assess the general morphology of neurons developing under *PMSC5* KD, we measured their dendritic arborization at DIV14. No significant differences were observed in *Psmc5* KD neurons compared to controls (Fig. 3o, p). We also measured synapse density of excitatory synapses (visualized using vGlut as excitatory presynaptic marker) and inhibitory synapses (visualized using vGat as inhibitory presynaptic marker). After 14 days, PSMC5 loss led to a significantly reduced number of vGlut positive puncta along primary dendrites (Fig. 3q, s), whereas the number of vGat positive puncta remained unchanged. (Fig. 3r, t). This indicates an excitation/inhibition (E/I) imbalance due to PSMC5 loss in neurons.

### Overexpression of *PSMC5* affects neuronal morphology in mouse

A subset of identified missense variants was assessed for their impact on neuronal morphology through ectopic expression in primary mouse hippocampal neurons. Neurite length of mouse hippocampal cells was significantly increased during the neurodevelopmental phase (DIV3) by expression of protein product PSMC5-WT compared to

empty vector control (Fig. 4a, b), suggesting a potential role of *Psmc5* in promoting neurite outgrowth. Overexpression of variants p.(Arg201Trp), p.(Pro320His) and p.(Arg325Trp) yielded the same effect as PSMC5-WT, suggesting that in this assay these variants do not alter the effect of PSMC5-WT expression. By contrast, overexpression of variants p.(Ala202Val), p.(Thr207Met), p.(Pro320Arg) and p.(Ala324Lys360del) (c.970-2A>G) showed reduced neurite length compared to PSMC5-WT (Fig. 4a,b); this effect, similar to the one induced by the empty vector, suggest that these variants have a loss of function (LoF) effect in the assay. Notably, no effect on arborization was induced by PSMC5-WT and none of the variants had impact on arborization compared to the empty vector control (Fig. 4c).

### The *PSMC5* variants associated with NDD do not equally impact the PSMC5/Rpt6 subunit steady-state expression and subsequent incorporation into 26S proteasomes

To analyze the ability of PSMC5/Rpt6 variant subunits to integrate 26S proteasome complexes, we ectopically expressed 13/26 identified alterations as N-terminally HA tagged-PSMC5 versions in SHSY-5Y neuroblastoma cells, as previously described[11].

While the plasmid-driven production of *PSMC5* transcripts was similar between the wild-type and the 13 variant cell lines (Supplementary Fig. 6a), steady-state protein expression levels of PSMC5/Rpt6 were differentially impacted by the variants: they were profoundly reduced by p.(Ala202Val) and p.(Pro320Arg), moderately by p.(Pro183Leu), p.(Arg201Trp), p.(Glu250Val)and p.(Arg325Trp), and mildly by p.(Gln221Arg), p.(Gly216Asp), p.(Arg258Trp), p.(Pro320His),

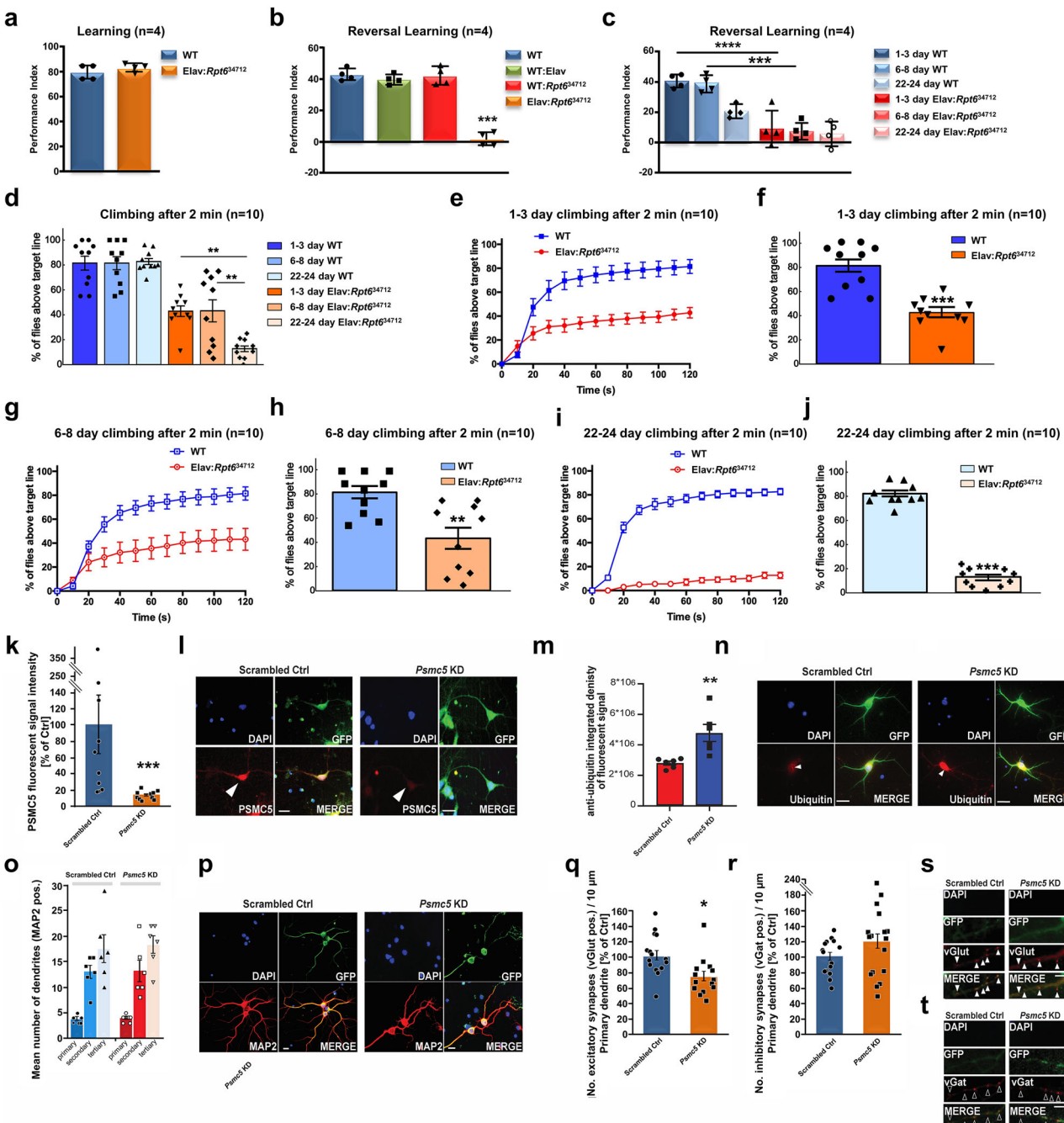

**Fig. 3 | *Psmc5* suppression impairs reversal learning in flies and alters synaptic balance in rat hippocampal neurons.** All data are presented as mean±SEM. **a–c** Pan-neuronal KD of *Rpt6* (*PSMC5*) in *Drosophila* had no effect on normal olfactory learning (two-sided Student's *t*-test; wild-type WT: 78 ± 2.9; KD(Elav:R*pt6*[34712]): 82 ± 1.7; *p* = 0.3059; *n* = 4 biological replicates of PI), but significantly impaired reversal learning (one-way ANOVA with Tukey post hoc; WT: 43 ± 1.9; WT:Elav: 40 ± 1.7; WT:*Rpt6*[34712]: 42 ± 3; KD(Elav:*Rpt6*[34712]): 1.5 ± 2; *p* < 0.0001, *n* = 4 PI), with age-persistent deficits. \*\*\**p* < 0.001; \*\*\*\**p* < 0.0001. **d–j** Climbing activity declined in Rpt6 KD flies over time: at 1–3 days (two-sided *t*-test; WT: 81.5% ± 5.8%; KD: 42.9% ± 4.4%; p < 0.0001; *n* = 10 biological replicates); at 6–8 days (two-sided *t*-test; WT: 81.5% ± 5.5%; KD: 43.1% ± 9.1%; *p* = 0.0021; *n* = 10 biological replicates); at 22–24 days (t two-sided *t*-test; WT: 82.7% ± 2.7%; KD: 12.7%±2.5%; *p* < 0.0001; *n* = 10 biological replicates). WT flies maintained performance across all ages. \*\**p* < 0.01; \*\*\**p* < 0.001. **k–n** Primary rat hippocampal neurons infected with lentiviral particles (GFP + scrambled or *Psmc5*-KD shRNA) showed: reduced PSMC5 protein in GFP positive (+) *Psmc5* KD neurons compared to scrambled controls

(two-sided Mann Whitney test; control: 100% ± 36.76%; KD: 12.71% ± 1.61%; *p* = 0.0002, *n* = 10 images analyzed per condition); and increased ubiquitin accumulation (two-sided Mann Whitney test; control: 2.8 × 10⁶ ± 1.3 × 10⁵; KD: 4.8 × 10⁶ ± 5.6 × 10⁵; *p* = 0.0022, *n* = 6). Scale bar = 20 μm. **o, p** No significant difference in dendritic branching was observed at DIV14 between *Psmc5* KD and control neurons (two-sided Mann-Whitney test). The two groups exhibited comparable numbers of primary dendrites (control: 3.67 ± 0.33; *Psmc5* KD: 3.83 ± 0.31; *p* = 0.749), secondary dendrites (control: 13.00 ± 1.37; KD: 13.17 ± 2.10; *p* = 0.873) and tertiary dendrites (control: 17.50 ± 6.69; KD: 18.33 ± 1.73; *p* = 0.575). *n* = 6 neurons per condition. MAP2 staining of infected neurons is shown in red. Scale bar: 20 μm. **q, r** Synaptic analysis revealed: Reduced vGlut⁺ excitatory puncta (two-sided Mann Whitney test; control: 1.2 ± 0.1; KD: 0.8 ± 0.1; *p* = 0.0045; *n* = 5) and stable vGat⁺ (two-sided Mann Whitney test; control: 0.8 ± 0.08; KD: 1.0 ± 0.2; *p* = 0.2988; *n* = 5) in KD neurons. **s, t** Decreased vGlut⁺ (filled arrows) and stable vGat⁺ (open arrows) signals in KD vs. control neurons. \**p* < 0.05; \*\**p* < 0.01. Scale bar = 5 μm.

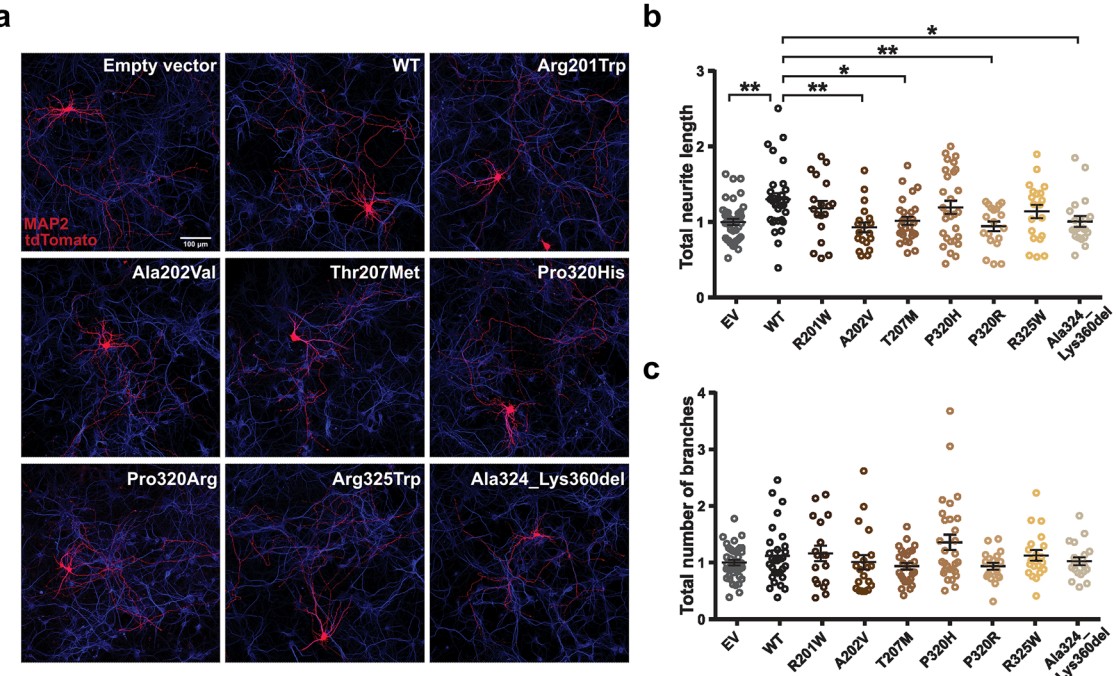

**Fig. 4 | *PSMC5* overexpression alters neuronal morphology. a** Representative images of primary hippocampal neurons transfected with empty vector (EV) control, wild-type PSMC5 (PSMC5-WT), or *PSMC5* variants. Transfected cells are labeled in red (tdTomato); MAP2 staining (blue) marks neuronal morphology. **b** Quantification of total neurite length shows: (i) a significant increase with PSMC5-WT (1.29 ± 0.08) compared to EV (1 ± 0.04; one-way ANOVA, $F = 3.296$, $p = 0.0014$; EV vs. PSMC5-WT, $p = 0.0046$, Dunnett's multiple comparisons test); (ii) Comparable effects for variants p.(Arg201Trp), p.(Pro320His), and p.(Arg325Trp) vs. PSMC5-WT: 1.18 ± 0.10, $p = 0.8$; 1.19 ± 0.09, $p = 0.8$; and 1.14 ± 0.09, $p = 0.5$; respectively; and (iii) a significant decrease for variants p.(Ala202Val), p.(Thr207Met), p.(Pro320Arg), and p.(Ala324Lys360del) compared to PSMC5-WT: 0.93 ± 0.07, $p = 0.0031$; 1.02 ± 0.06, $p = 0.02$; 1.19 ± 0.09; $p = 0.006$; and 1.00 ± 0.07, $p = 0.03$, respectively, with effects indistinguishable from EV ($p = 0.4$, $p = 0.9$, $p = 0.9$, and $p = 0.9$, respectively). **c** Sholl analysis revealed no significant differences in dendritic arborization across conditions (one-way ANOVA, $F = 2.179$, $p = 0.03$; EV vs. PSMC5-WT: 1.12 ± 0.09, $p = 0.9$; PSMC5-WT vs. p.(Arg201Trp): 1.16 ± 0.14, $p = 0.9$; PSMC5-WT vs. p.(Ala202Val): 1.00 ± 0.12, $p = 0.9$; PSMC5-WT vs. p.(Thr207Met), 0.94 ± 0.06, $p = 0.6$; PSMC5-WT vs. p.(Pro320His), 1.36 ± 0.14, $p = 0.3$; PSMC5-WT vs. p.(Pro320Arg), 0.94 ± 0.06, $p = 0.9$; PSMC5-WT vs. p.(Arg 325Trp), 1.13 ± 0.09, $p = 0.9$; PSMC5-WT vs. p.(Ala324Lys360del), 1.03 ± 0.07 $p = 0.9$; Dunnet's multiple comparison test). Error bars indicate SEM; n (number of neurons traced): EV = 40, PSMC5-WT = 30, p.(Arg201Trp) = 18 p.(Ala202Val) = 20, p.(Thr207Met) = 27, p.(Pro320Arg) = 29, p.(Pro320Arg) = 19, p.(Arg325Trp) = 20 and p.(Ala324-Lys360del) = 20. \*$p < 0.05$; \*\*$p < 0.01$. Scale bar: 100 μm.

and p.(Met368Thr), whereas they were unaltered by p.(Thr207Met) and p.(Asp394Glufs\*2) (Supplementary Fig. 6b). Overall, analyses of the 13 *PSMC5* variants examined revealed no significant impact on in vitro PSMC5/Rpt6 abundance when compared to control (Supplementary Fig. 6b, right panel), indicating that haploinsufficiency is unlikely the main driver of variant pathogenicity.

We also observed distinct effects of the 13 *PSMC5* variants on proteasome assembly in this in vitro assay. Notably, p.(Ala202Val) prevented PSMC5 incorporation into 19S, 26S, and 30S complexes (Supplementary Fig. 6c), due to the instability of the HA-PSMC5/Rpt6 full-length protein (Supplementary Fig. 6c). Conversely, despite their stable expression in SH-SY5Y cells (Supplementary Fig. 6b), p.(Met368Thr) and p.(Asp394Glufs\*2) accumulated in 19S precursors and fully assembled 19S particles without forming 26S and/or 30S proteasome complexes (Supplementary Fig. 6c). Unlike variants p.(Gly216Asp), p.(Gln221Arg), p.(Arg201Trp) and p.(Arg258Trp) which showed unchanged incorporation efficiency compared to their wild-type counterpart, p.(Pro183Leu), p.(Glu250Val), p.(Pro320His) and p.(Pro320Arg) only minimally assembled into mature proteasomes, (Supplementary Fig. 6c). Surprisingly, p.(Thr207Met) had a higher propensity to integrate 19S-capped proteasomes than the wild-type. Densitometric analysis showed that the 13 PSMC5 variants studied had similar incorporation into 19S, 26S, and/or 30S complexes compared to their wild-type counterparts (Supplementary Fig. 6c, lower panel). These findings underscore diverse effects of these variants on subunit expression, protein level stability, and incorporation into mature 26S/30S proteasome complexes.

## Most *PSMC5* variants do not lead to haploinsufficiency but result in severe proteasome assembly defects

In T cells from affected subjects, none of the *PSMC5* variants reduced PSMC5/Rpt6 levels compared to controls, except p.(Pro320Arg) (Fig. 5a). These levels remained unchanged with p.(Arg201Trp), p.(Glu250Val) or p.(Arg325Trp), contrary to in vitro findings in SHSY-5Y cells that had suggested variant-induced protein instability (Supplementary Fig. 6b). This discrepancy may reflect full compensation by the wild-type allele, countering haploinsufficiency as a cause of the disorder. It could also imply a greater turnover of PSMC5 in SH-SY5Y cells attributable to protein-level dosage compensation of the excess of subunits produced following *PSMC5* overexpression[19]. Notably, while steady-state PSMC5/Rpt6 levels were similar to controls, T cells with *PSMC5* variants exhibited additional lower-migrating PSMC5/Rpt6 species, potentially due to dysregulated *PSMC5* transcript splicing (Fig. 5a, lower panel).

In line with in vitro findings, T cells of subjects S25 [p.(Pro320Arg)], S36 [p.(Arg325Trp)], and S37 [p.(Arg325Trp)] exhibited reduced incorporation of PSMC5/Rpt6 into 26S and 30S proteasome complexes, indicated by reduced PSMC5/Rpt6 staining in 19S-capped proteasomes (Fig. 5b), and confirmed by mass spectrometry-based interactome analysis (Fig. 5c, d). However, in vivo/in vitro discrepancies were observed for other variants: (i) p.(Arg201Trp) and p.(Gln221Arg) reduced PSMC5/Rpt6 and PSMA6/α6 incorporation in T cells from S14 (Fig. 5b). Similar to the unique PRAAS case involving a *PSMC5* variant reported so far[20], an oligogenic mechanism could underlie the disorder of S14, who harbors a second proteasomal variant, p.(Arg90\*) in *PSMD11*, reported pathogenic elsewhere[12]. This

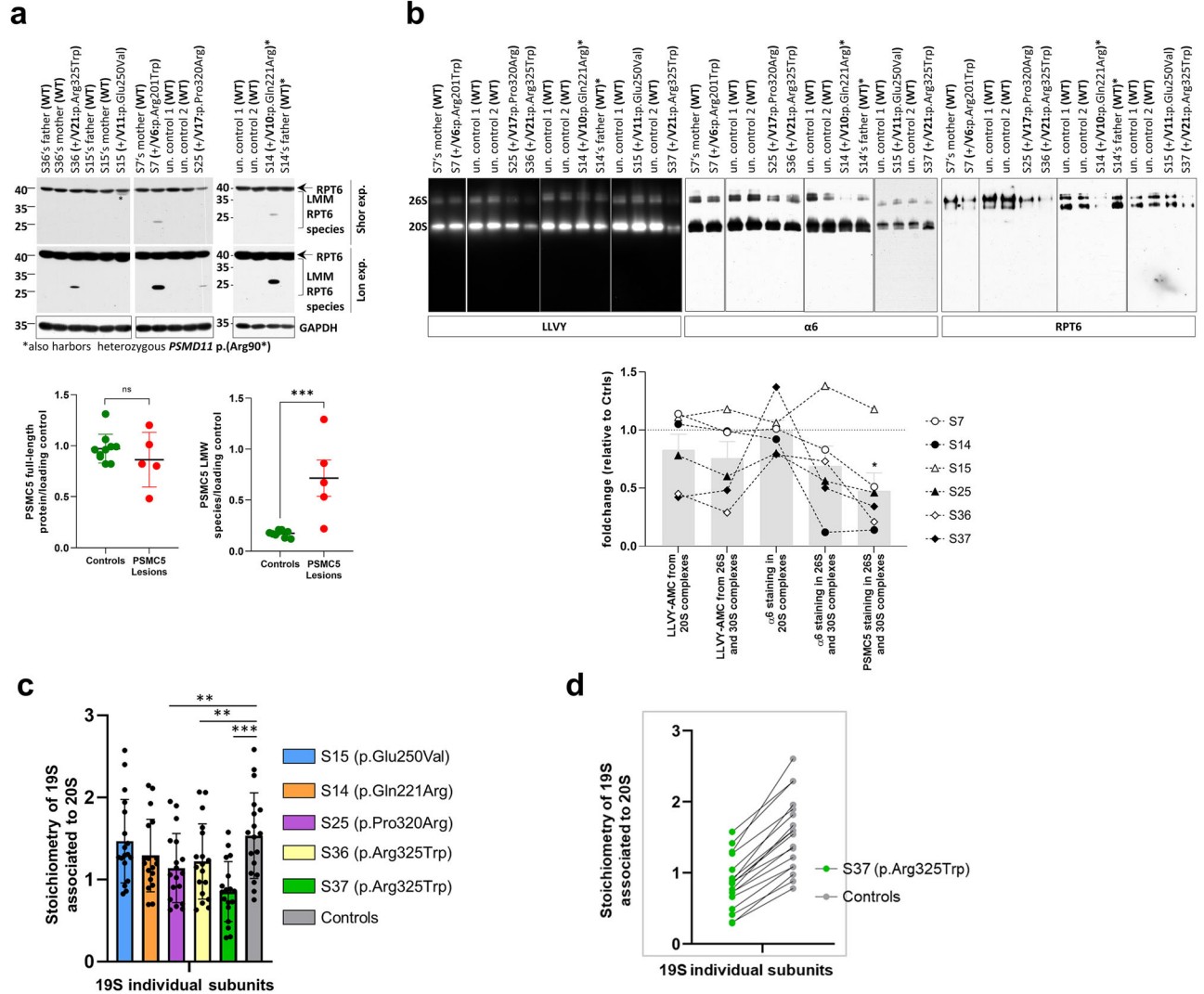

**Fig. 5 | Altered proteasome expression and activity in T cells from NDD subjects harboring *PSMC5* variants. a** Western blot analysis of T cells expanded from PBMCs of NDD subjects (S7, S14, S15, S25, S36), related controls (parents), and unrelated healthy donors. Lysates were probed with anti-PSMC5/RPT6 and GAPDH antibodies. Full-length PSMC5/RPT6 (~45 kDa; arrow) was detected in all samples. Additional lower molecular weight (LMM) bands (~30 kDa; brackets) were observed in S7, S14, S25, and S36, while S15 showed a distinct band (~40 kDa; asterisk). Lower panel: Densitometric quantification of full-length and LMM species normalized to GAPDH. Data are shown as mean±SD of biological replicates for controls ($n = 9$) and patients ($n = 5$); two-tailed Mann–Whitney $U$ test (***$p = 0.0007$). **b** Native-PAGE analysis of resting T cells from NDD subjects (S7, S14, S15, S25, S36, S37; $n = 6$), related controls, and healthy donors. Non-denatured lysates were prepared with TSDG buffer and probed with anti-α6 and anti-PSMC5/RPT6 antibodies.

Proteasome complexes (20S, 26S) were visualized by in-gel activity assay using the fluorogenic peptide Suc-LLVY-AMC. Lower panel: Quantification of LLVY-AMC fluorescence and α6/RPT6 signals in 30S/26S complexes, shown as fold change relative to controls (normalized to 1). Data represent mean±SD of biological replicates ($n = 6$); two-tailed Mann–Whitney $U$ test (*$p = 0.0476$). **c** Bottom-up mass spectrometry (BU-MS) analysis of 20S-associated 19S subunits in immunopurified proteasomes from five patient and six control T cells ($n = 11$). Each point represents the intensity of an individual 19S subunit; statistical analysis by one-way ANOVA with repeated measures, Geisser–Greenhouse correction, and Šidák's multiple comparison (***$p < 0.001$, **$p < 0.01$). Data are presented as mean values ± SD of biological replicates for controls ($1.53 \pm 0.5$; $n = 6$) and patients (S15 $1.47 \pm 0.5$; S14 $1.29 \pm 0.4$; S25 $1.14 \pm 0.4$; S36 $1.22 \pm 0.5$; S37 $0.85 \pm 0.4$; $n = 5$). **d** Individual 19S subunit intensities for subject S37 and matched controls from (**c**).

variant is inherited from a moderately affected father, in whom it reduced 30S proteasomal complexes and α6 staining intensity (Fig. 5b) (ii) p.(Glu250Val) did not impair PSMC5/Rpt6 assembly into 26S/30S proteasomal complexes in S15's T cells (Fig. 5b) but caused significant disruption in SHSY-5Y cells (Supplementary Fig. 6c). Overall, *PSMC5* variants show varied effects on proteasome stability and incorporation.

**PSMC5 variants profoundly disrupt protein homeostasis and lipid metabolism, and activate mitophagy in T cells from NDD subjects**

To assess how *PSMC5* variants affect protein homeostasis, we examined whether T cells from individuals with these variants tend to form protein aggregates. We observed increased aggresome formation in

samples from S7/14/15/25/36 (Fig. 6a), along with the accumulation of high molecular weight (HMW) ubiquitin-modified proteins in samples from S1/7/14/15/25/36 (Fig. 6b). These findings demonstrate the loss-of-function nature of *PSMC5* variants, which prevent T cells from coping with proteotoxic stress.

To explore the cellular repercussions of proteasome dysfunction, we conducted two separate series of proteomic analyses on subject-derived T cells. Both analyses revealed an enrichment in deregulated mitochondrial proteins (Fig. 7a and Supplementary Fig. 7b). Up-regulated proteins included those involved in membrane structure (MT-ATP8, MTX2, MTCH2) and mitochondrial protein homeostasis and assembly (DNLZ, COX19, GTPBP6), while down-regulated proteins were associated with maintaining mitochondrial integrity under

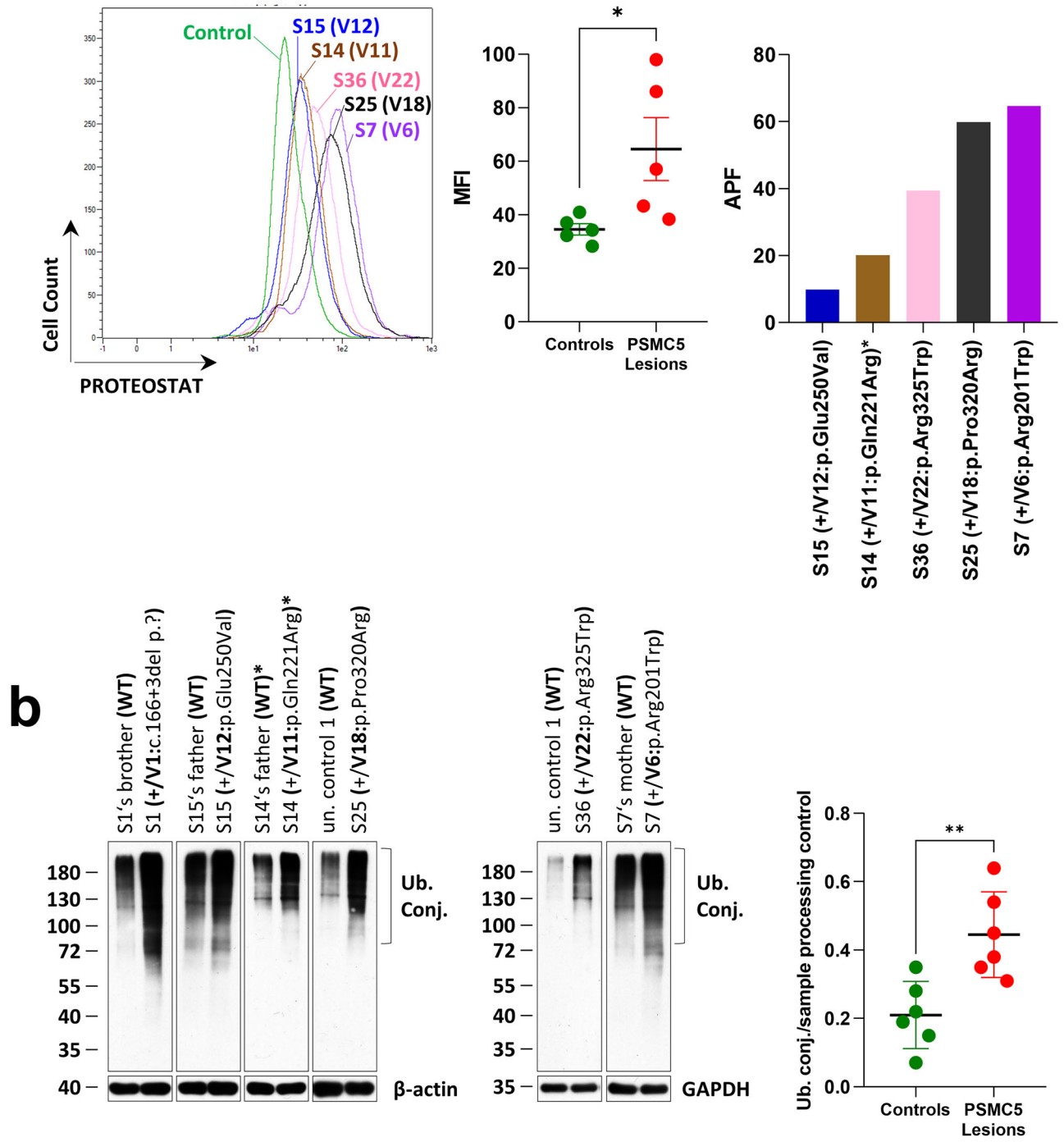

**Fig. 6 | *PSMC5* variants disrupt protein homeostasis in T cells. a** T cells expanded from PBMCs of NDD subjects (S7, S14, S15, S25, S36; *n* = 5) were stained with 1 μM of the PROTEOSTAT® dye and analyzed by flow cytometry using the B3 (PerCP-Vio 700) channel. PROTEOSTAT® selectively intercalates into the cross-β structures of misfolded and aggregated proteins. All T cells harboring *PSMC5* variants exhibited elevated aggresome formation, as indicated by increased PROTEOSTAT® fluorescence intensity. Left panel: representative histogram overlay comparing the six patient samples to a healthy control. Middle panel: mean fluorescence intensity (MFI) values of activity-based probe (ABP)-treated T cells are shown as mean ± SEM of biological replicates for patient (*PSMC5* variant, 64.53 ± 11.77, *n* = 5) and control (34.58 ± 2.148, *n* = 5) groups. *\*p* = 0.0159, two-tailed Mann–Whitney *U* test. Right

panel: aggresome propensity factor (APF) calculated for each patient as the percentage difference in MFI between MG-132–treated and untreated cells, normalized to the MG-132–treated condition. **b** T cells from NDD subjects (S1, S7, S14, S15, S25, S36; *n* = 6), their relatives (father, mother and/or brother), and unrelated healthy donors (controls, *n* = 6) were subjected to RIPA-based protein extraction followed by SDS-PAGE and western blotting using antibodies against ubiquitin, β-actin, and GAPDH (sample processing controls). Right panel: densitometric quantification of ubiquitin normalized to β-actin. Data are presented as mean ± SD for control (*n* = 6) and patient (*PSMC5* variant, *n* = 6) groups. *\*\*p* = 0.0047, two-tailed Mann–Whitney *U* test.

oxidative stress or hypoxia (BOLA1, BCL2L2, AK4, GLRX5). Measurement of mitophagy rates using the pH-sensitive MtPhagy dye showed that T cells with *PSMC5* variants (S1/7/14/15/25/37) had a higher percentage of lysosomal-targeted mitochondria than controls (Fig. 7e), confirming that proteasome dysfunction is linked to increased mitochondrial removal and damage.

We also noted an enrichment of down-regulated proteins involved in the early stages of ribosome biogenesis, particularly rRNA processing and 90 s preribosomes assembly. This was especially pronounced in the second proteomic series (e.g., MAK16, UTP11, UTP25, PUM3, WDR2, DDX18; Fig. 7b) but was also evident, though less marked, in the first series (e.g., WDR36, TBL3, EBP2; Supplementary Table 3d, e). Transcriptomics of T cells from affected subjects further supported an impairment of ribosome biogenesis (Supplementary Fig. 8a).

Since glycerophospholipid metabolism was also found significantly deregulated (Fig. 7b), we performed T cell lipididomics, which showed massively altered lipid distribution in samples from subjects with *PSMC5* variants (Fig. 7c). Levels of membrane phospholipids such as phosphatidylserine (PS), phosphatidylethanolamine (PE), phosphatidylcholine (PC), or di- (DG) or tri-acylglycerols (TG), were significantly decreased in patients, whereas cholesterolesters were increased by 50% (Fig. 7d).

### Individuals harboring PSMC5 variants display sterile type I IFN responses predominantly triggered by the ISR

Referring to the invariably dysregulated type I interferon (IFN) signaling reported in both PRAAS and NDPs, we performed a targeted transcriptomic analysis on a NanoString panel of autoimmune genes that revealed a strongly elevated type I IFN signature in patients compared to controls (Fig. 8a) also confirmed by T cells transcriptomics (Fig. 8b and Supplementary Fig. 8b).

To further corroborate the association between *PSMC5* variants and a type I IFN gene signature, we calculated the IFN scores of T cells from S1/14/15/25/36/37 and unaffected controls based on expression of seven IFN-stimulated genes (ISGs: *IFIT1*, *IFI27*, *IFI44*, *IFI44L*, *ISG15*, *MX1* and *RSAD2*). All individuals harboring *PSMC5* variants showed elevated expression of ISGs compared to controls (Supplementary Fig. 9), resulting in high IFN scores (Fig. 8c). Intriguingly, the father of S14, carrying the heterozygous *PSMD11* variant p.(Arg90*) (Supplementary Table 2), exhibited a positive type I IFN score (Fig. 8c), suggesting that proteasome loss-of-function variants consistently lead to type I IFN gene signatures, even in mildly affected subjects.

The established role of protein kinase R (PKR) in initiating sterile autoinflammatory response to proteasome dysfunction[21] prompted us to assess its contribution to the initiation of type I IFN responses in T cells from subjects with *PSMC5* variant by using C16, a specific PKR inhibitor. The IFN scores from T cells treated with C16 were significantly lower in T cells treated with C16 than in the same T cells exposed to DMSO (controls), confirming PKR's role in spontaneous ISG induction in this disorder (Fig. 8d, Supplementary Fig. 10). A comparable reduction was observed upon H-151 treatment (Fig. 8d, Supplementary Fig. 10), underscoring an additional contribution from the cGAS-STING pathway, which senses host-derived DNA in this process that may originate from damaged mitochondria in this setting. Of note, the IFN scores exhibited a dramatic decrease upon treatment with JAK inhibitor baricitinib (Fig. 7c, Supplementary Fig. 8), confirming the autocrine/paracrine nature of this phenomenon. Conversely, blocking the UPR with the IRE1 inhibitor 4μ8C did not yield any significant effects (Fig. S8). Inhibition of another ISR kinase, GCN2, with A92 led to a significant reduction in IFN scores for all subjects, emphasizing the key role of ISR in the generation of type I IFN.

### The recurrent *PSMC5* heterozygous variant c.973C>T p.(Arg325Trp) perturbs the differentiation of induced pluripotent stem cells (iPSCs) into neural progenitor cells

Given the syndromic and multisystem nature of the condition, we reasoned that proteasome dysfunction caused by *PSMC5* variants might disrupt early developmental processes. To address this point, we evaluated the impact of variant c.973C>T p.(Arg325Trp), generally associated with a multi-systemic severe phenotype on the differentiation potential of iPSCs. To this end, the three variant iPSC clones PSMC5(+/p.Arg325Trp) edited by CRISPR/Cas9 and their three isogenic WT controls PSMC5(+/p.Arg325Trp) were used as precursors to generate the three germ layers, and neural progenitor cells (NPCs) (Fig. 9a).

Differentiation of PSMC5(+/+) and PSMC5(+/p.Arg325Trp) iPSCs into the three germ layers resulted in the successful generation of ectodermal, mesodermal, and endodermal cells expressing comparable levels of stage-specific differentiation markers: SOX1 (ectoderm), HAND1 and TBXT (mesoderm), and FOXA2 and SOX17 (endoderm), as assessed by RT-qPCR (Fig. 9b). Notably, a slight but non-significant decrease in the ectodermal marker PAX6 was observed in PSMC5(+/p.Arg325Trp) cells compared to PSMC5(+/+) suggesting minimal impact of variant *PSMC5* c.973C>T p.(Arg325Trp) on the formation of the three layers. However, a significant down-regulation of *HES1* was observed by transcriptomics in PSMC5(+/p.Arg325Trp) ectodermal cells compared to PSMC5(+/+) cells (Fig. 9c and Supplementary Fig. 11). Transcriptomics also showed an increased expression of *MSMO1* (Fig. 9c, d), which indicates a likely deregulated cholesterol metabolism (Fig. 9d and Supplementary Fig. 11).

Considering the neurodevelopmental abnormalities observed in all subjects (Fig. 2a), we also assessed the impact of *PSMC5* variants on neuronal processes, particularly neurogenesis, by differentiating PSMC5(+/+) and PSMC5(p.Arg325Trp) iPSCs into neural progenitor cells (NPCs). Nearly all day-28 NPCs derived from PSMC5(+/+) iPSCs expressed the neuronal markers Nestin and SOX2 (Fig. 9e, f). In contrast, a significant portion of NPCs derived from PSMC5(+/p.Arg325Trp) iPSCs failed to express these markers, as determined by fluorescence microscopy. Indeed, a thorough analysis of the microscopy images revealed that the proportion of SOX2-positive cells was 10% lower, and the proportion of Nestin-positive cells was 7% lower in PSMC5(+/p.Arg325Trp) NPCs compared to wild-type NPCs (Fig. 9e, g). These results confirm that the *PSMC5* c.973C>T p.Arg325Trp variant is associated with impaired neurogenesis. Altogether, these data highlight the detrimental role of proteasome variants in early development, specifically by impairing the genesis of NPCs. These findings establish a clear cause-to-effect relationship between *PSMC5* variants and the observed clinical phenotype.

## Discussion

The growing recognition of rare syndromic NDPs highlights the vulnerability of genes encoding 19S proteasome subunits like *PSMC5* to genomic lesions, as observed in various CNS disorders[9–12]. The pivotal role of *PSMC5* in early human developmental processes had been strongly suggested by a significant enrichment of de novo variants in individuals with developmental disorders in a large international cohort study[22]. This is further confirmed in our study by the multisystemic anomalies ascertained in subjects with *PSMC5* variants (Fig. 2a and Supplementary Table 2) and underlines previous findings in mice and plants[23,24]. Furthermore, our investigation of an iPSC-based model demonstrates that the variant p.(Arg325Trp) does not impair the generation of the three germ layers (Fig. 9b), but affects developmental stages (Fig. 9e–g). Given the critical role of proteasomes in maintaining pluripotency[25] and the higher proteasome activity in iPSCs[26], this observation suggests that iPSCs and germ layer cells may compensate for the detrimental effects of *PSMC5* variant p.(Arg325Trp) by monoallelic expression of the wild-type *PSMC5* allele and/or an enhanced capacity to selectively

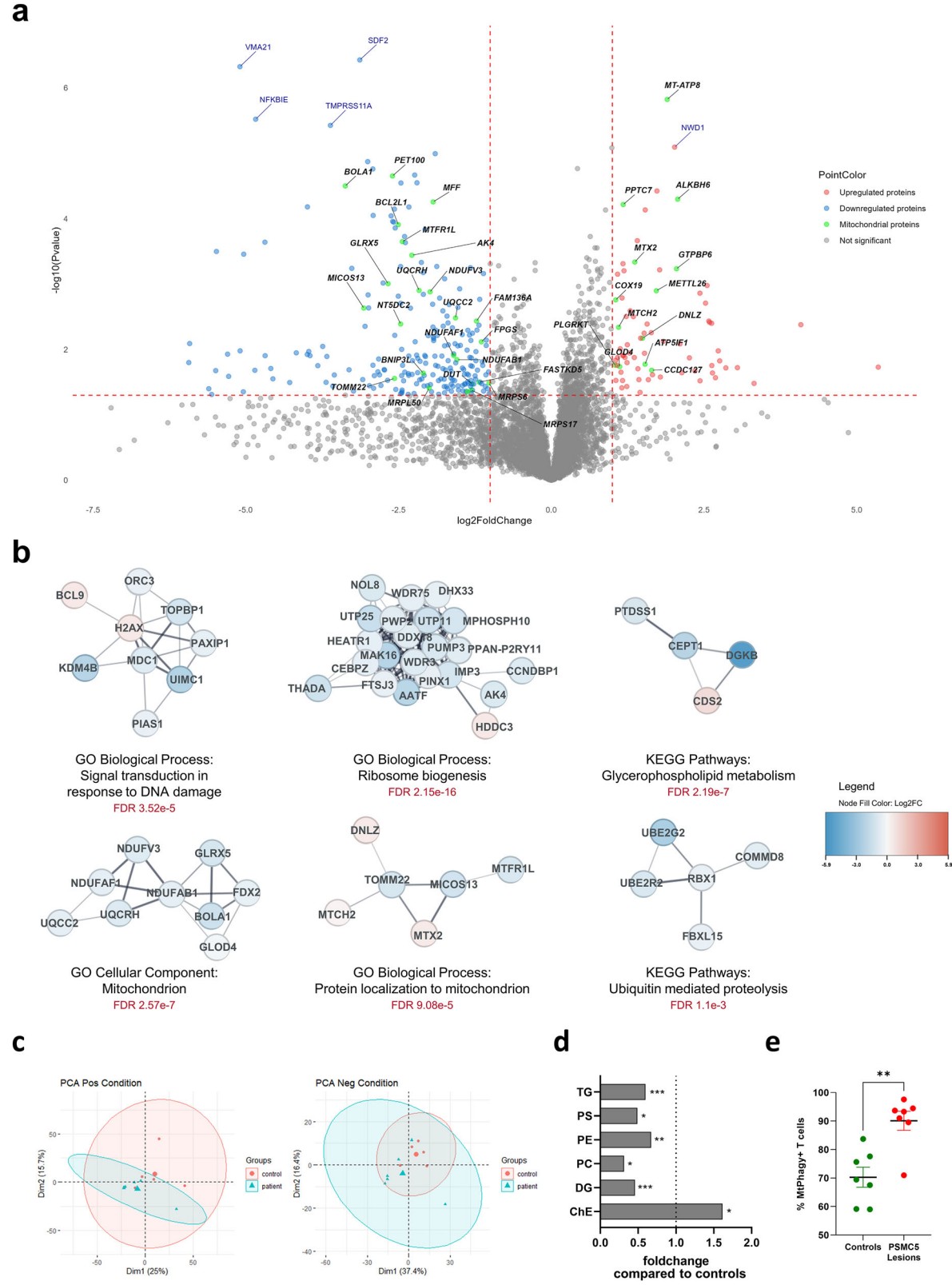

eliminate defective proteasome assembly intermediates[27]. Tolerance to *PSMC5* variants progressively diminishes during differentiation, the adverse effects of the variant becoming evident at later developmental stages, notably during the genesis of NPCs (Fig. 9e–g). The particular vulnerability of NPCs to proteasome dysfunction could partly derive from the down-regulation of *HES1* observed in iPSC-derived ectodermal cells (Fig. 9c). HES1 acts as a

transcriptional repressor important for the determination of neural progenitor cell proliferation and fate[28,29]. Although it is not yet clear whether differentiated tissues derived from mesodermal and/or endodermal lineages share this susceptibility to proteasome dysfunction with NPCs, these results may explain the predominance of a neurodevelopmental phenotype in affected individuals with *PSMC5* variants (Fig. 2a–d, Supplementary Table 2).

**Fig. 7 | *PSMC5* variants impair mitochondrial homeostasis, ribosome biogenesis and lipid metabolism in T cells. a** Volcano plot of differentially expressed proteins in T cells from five individuals (four samples: three separate samples for S14, S15, S25, plus pooled sample from S36 and S37) versus nine controls, identified by mass spectrometry (*n* = 13 samples; *p* ≤ 0.05; log$_2$FC ≥ 1 or ≤ −1; Student's *t*-test). Red and blue dots indicate enriched or depleted pathways; green dots highlight mitochondrial proteins (MitoCarta 3.0)[69]. **b** Proteomics: Network clustering and pathway analysis of dysregulated proteins (*p* ≤ 0.05; −2 ≤ FC ≤ 2; Student's *t*-test) using GO, KEGG, and STRING databases in Cytoscape (STRING *Homo sapiens* background). **c** Untargeted lipidomics of T cells from patients (S1, S7, S14, S15, S23, S25, S37; *n* = 7) and matched controls (parents and/or siblings; *n* = 6) revealed distinct

lipid profiles by PCA in positive and negative ion modes (patients: blue; controls: red). **d** Fold-change analysis of lipid classes showed significant alterations in patient T cells (*p* < 0.05, **p* < 0.01, ***p* < 0.001; two-tailed Student's *t*-test). **e** Mitophagy was assessed by flow cytometry in T cells from patients (S1, S7, S14, S15, S25, S36, S37) and healthy donors (related and unrelated) using Mtphagy dye (100 nM, overnight; B4 channel, PE-Vio 770). Percentages of mitophagy-positive cells were significantly increased in patients (**p* = 0.002, two-tailed Mann-Whitney *U* test). Data are presented as the mean percentage of mitophagy-positive cells ±SEM of biological replicates for controls (90.09 ± 3.330, *n* = 7) and patients (70.29 ± 3.525, *n* = 7).

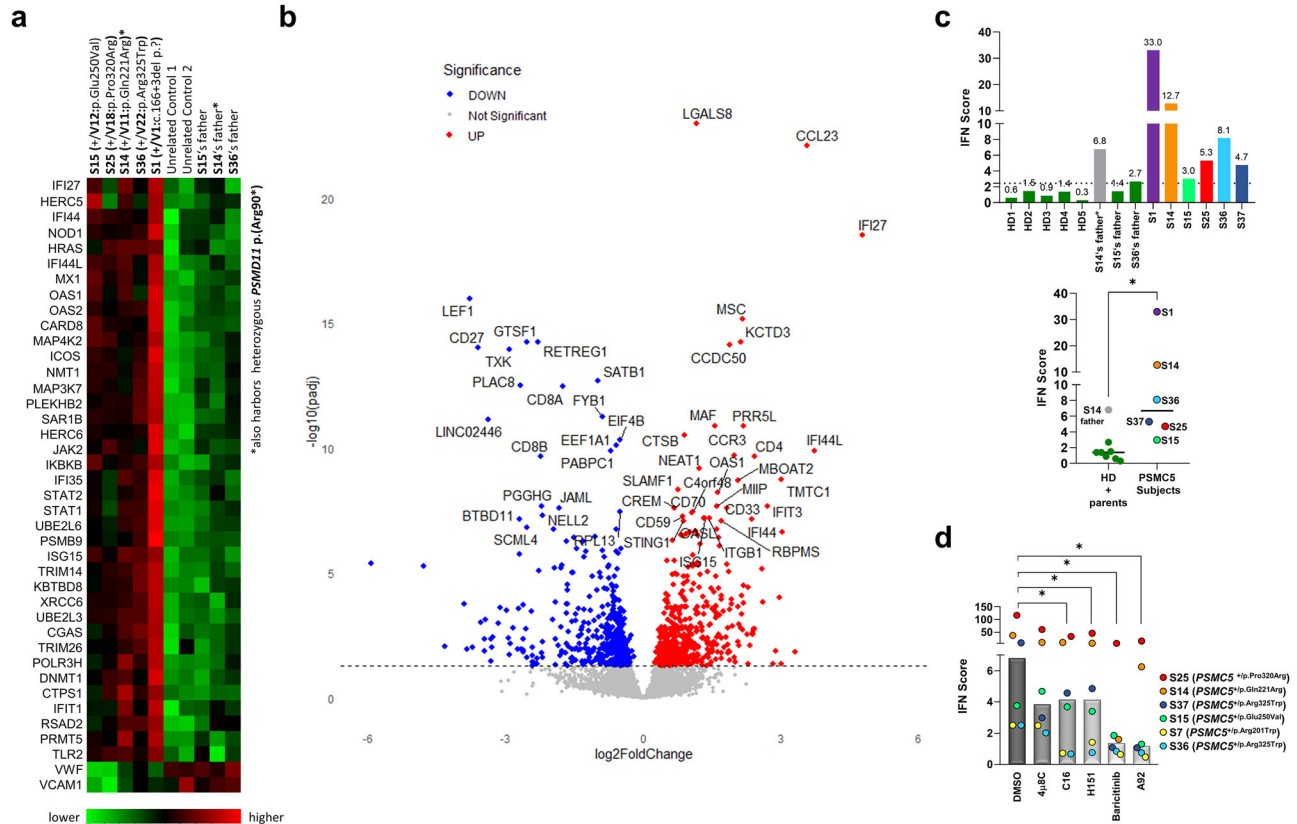

**Fig. 8 | *PSMC5* variants trigger a spontaneous type I IFN response and distinct immune signature in T cells. a** Heatmap of immune-related gene expression in T cells from five NDD subjects (S1, S14, S15, S25, S36; *n* = 5) versus five healthy controls (probands' fathers and unrelated donors; *n* = 5), profiled using NanoString nCounter® (700-gene panel)[11]. Forty genes were differentially expressed, predominantly upregulated type I interferon-stimulated genes (ISGs), including *STAT1*, *STAT2*, *UBE2L6*, and *HERC5*. **b** Volcano plot from 3′ digital gene expression profiling showing 507 upregulated and 452 downregulated genes in *PSMC5* variant carriers versus controls (padj≤0.05). Red and blue dots indicate upregulated and

downregulated genes, respectively. **c** RT-qPCR quantification of seven ISGs (*IFIT1*, *IFI27*, *IFI44*, *IFI44L*, *ISG15*, *MX1*, *RSAD2*) in T cells from six patients (*n* = 6; S1, S14, S15, S25, S36, S37), healthy donors (HD), and eight related controls (parents; *n* = 8). Data represent biological replicates and type I IFN scores were calculated as median fold-change relative to a calibrator (*p* = 0.0043, two-tailed Mann-Whitney *U* test). **d** T cells from six patients (S1, S14, S15, S25, S36, S37) were treated for 8 h with DMSO (vehicle), 4µ8C (100 µM), C16 (3 µM), H-151 (2 µM), baricitinib (1 µM) or A92 (10 µM) prior to ISG quantification by RT-qPCR. Data represent biological replicates and IFN scores under each condition (*p* = 0.0313, Wilcoxon test).

The difficulty for differentiated cells to cope with proteasome deficiency was demonstrated by the accumulation of polyubiquitinated proteins caused by *Psmc5* KD in primary rat hippocampal neurons (Fig. 3s, t) and *PSMC5* variants in human subjects' T cells (Fig. 6a, b). The mechanism leading to the loss of excitatory synapses—and consequently to a reduced E/I ratio—associated with proteostatic perturbation in *Psmc5*-KD neurons (Fig. 3q, r) may involve the proteasome-independent function of free 19S particles in regulating excitatory synapse transmissions in cortical neurons[30]. Similar imbalances in E/I ratio are associated with impaired decision-making[31], Alzheimer's disease[32] or cognitive deficits in conditions like ASD and schizophrenia[33]. The comparison with neuropsychiatric disorders can be extended to the neuritogenesis impairment caused by *PSMC5*

variants (Fig. 4). These findings are consistent with the role of Rpt6 on dendritic spine outgrowth in rat hippocampal cells[34], also induced by *AUTS2* alterations in addition to an E/I imbalance[35].

The interaction between the PSMC5 subunit and cognitive performance is substantiated by the evaluation of the *Drosophila Rpt6(PSMC5)*-KD model, which highlighted a specific pattern of brain functional abnormalities such as impairment of reversal learning without compromising memory (Fig. 3a–c). These findings align with observations from the paralogous gene *Rpt5(PSMC3)* in flies[11] or in rodents showing the involvement of *Psmc5* in behavioral plasticity[36,37], through regulation of chromatin remodeling and gene expression in the nucleus accumbens. The impact of PSMC5 on reverse learning becomes also evident by the modulation of GABAergic signaling[38], a

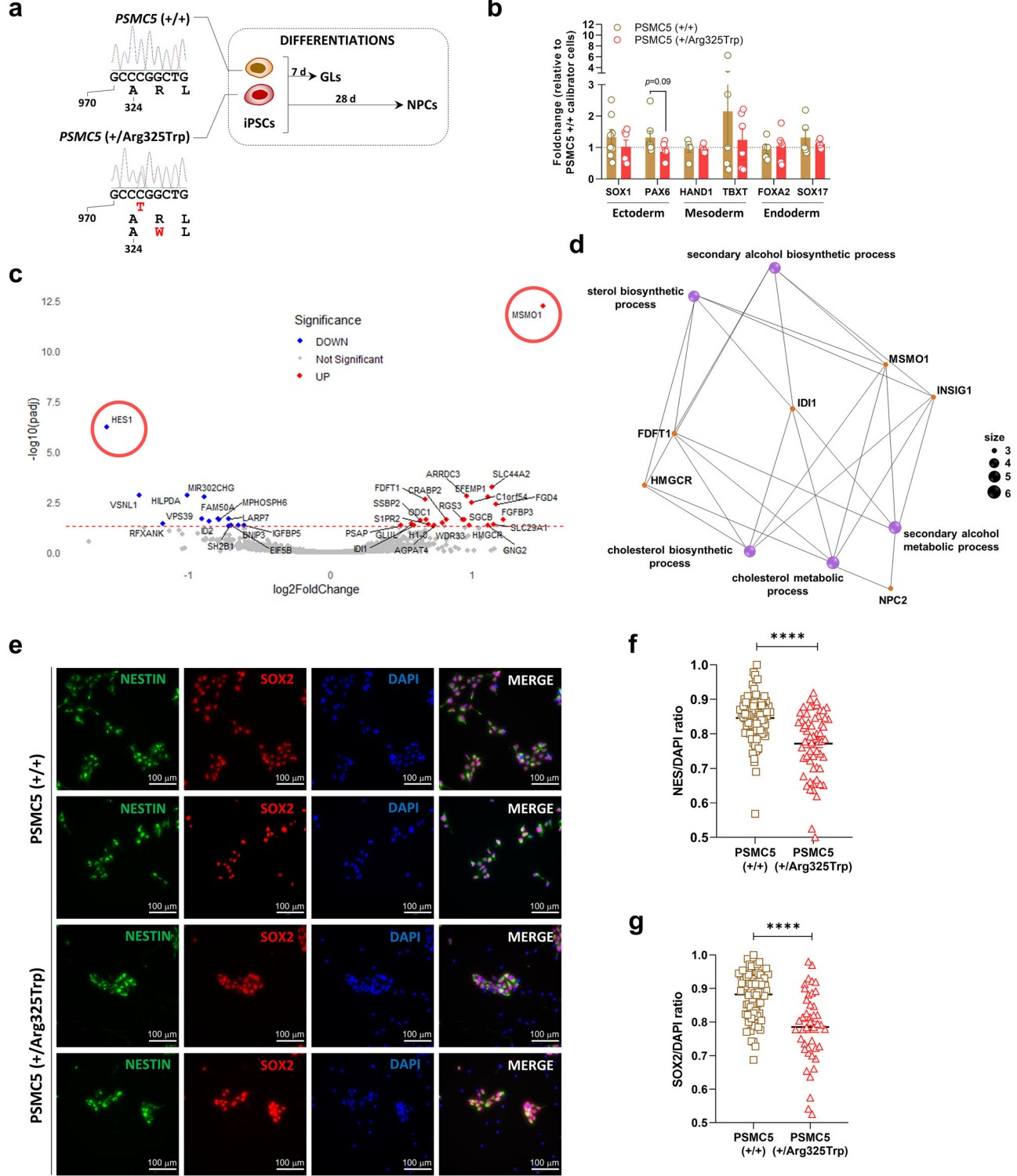

**Fig. 9 | *PSMC5* variant c.973C>T p.(Arg325Trp) impairs iPSC differentiation into germ layers and neural progenitors. a** Schematic of ASE9211 iPSC clones edited via CRISPR/Cas9 to introduce the heterozygous p.Arg325Trp variant. Wild-type clones (ASE9211, C10, B10, E5) and mutant clones (G1, G8, H1) were differentiated into day-7 germ layer (GL) cells and day-28 neural progenitor cells (NPCs). **b** RT-qPCR of lineage-specific markers in GL cells: ectoderm (*SOX1*, *PAX6*), mesoderm (*HAND1*, *TBXT*), and endoderm (*FOXA2*, *SOX17*), normalized to RPLP13 and expressed relative to wild-type clone C10. Data represent biological replicates from six independent experiments for mutant cells. For wild-type cells, sample sizes were n = 4 for *HAND1*, *TBXT*, *FOXA2*, and *SOX17*; n = 6 for *PAX6*; and n = 7 for *SOX1*

(two-tailed unpaired *t*-test). **c** Volcano plot of differentially expressed genes in ectodermal cells from wild-type and mutant clones (padj≤0.05): 23 upregulated (red), 14 downregulated (blue). **d** Cnetplot of gene set enrichment analysis showing upregulated genes (orange) and enriched pathways (purple), notably cholesterol and sterol metabolism. **e** Immunofluorescence of NPCs stained for nestin (green), SOX2 (red), and DAPI (blue), comparing wild-type (ASE9211, E5) and mutant (G1, G8) clones. Quantification of nestin⁺ (**f**) and SOX2⁺ (**g**) cells relative to DAPI⁺ nuclei in NPCs from wild-type and mutant clones (≥20 images per line analyzed using Fiji software; mean±SEM; ****p < 0.0001, two-tailed unpaired *t*-test).

key component of behavioral flexibility[39]. Furthermore, the Rpt6-KD *Drosophila* model showing that climbing dramatically deteriorates at about ¾ of the fly lifespan (Fig. 3d) may provide a clue towards the possible clinical evolution of *PSMC5*-related manifestations in humans. In the series, this finding resonates with cases of developmental regression in children (S22/23/29) and cerebral atrophy or progressive MRI anomalies in adults (S9/16/36), suggesting a possible association with neurodegenerative processes. This stresses the importance of the currently limited follow-up on clinical progression.

Despite the prevailing neurodevelopmental impact of *PSMC5* dysfunction, virtually all body cells are affected. The trend towards increased viral and interferon responses already noted in PSMC5$^{(+/p.Arg325Trp)}$ iPSC-derived mesodermal cells (Supplementary Fig. 12) suggested that the vulnerability to proteasomal dysfunction could extend to the hematopoietic lineage derived from the mesoderm. This was validated in T cells, which therefore served as an accessible in vitro cell model enabling to profile a characteristic phenotype of cells with *PSMC5* variants (Figs. 5 and 6), increased mitophagy (Fig. 7e, and Supplementary Fig. 7b), and a type I IFN gene signature (Fig. 8c and Supplementary Fig. 9).

Proteasome impairment, activation of proteotoxic stress and dysregulation of type I IFN signaling are also hallmarks of PRAAS, mainly caused by variants in 20S proteasome subunits[8,40] (Supplementary Fig. 1). Although individuals with *PSMC5*-associated NDD lack the pronounced inflammatory symptoms of PRAAS[40], a definite clinical overlap exists. Some of the patients in our cohort showed dysregulation of the hematopoietic system, including recurrent infections, periodic fever or thrombocytopenia (Supplementary Table 2). Similar to the Aicardi-Goutières-Syndrome, a prototypical interferonopathy with CNS manifestation, a substantial number of PRAAS patients in turn present with intellectual disability and global developmental delay indicating overlaps between PRAAS and NDPs[41]. The link between dysregulation of type I IFN signaling and proteotoxic stress in proteasompathies becomes evident through the proteotoxic stress markers PKR and GCN2. These two sensors of the ISR belong to the eIF2α kinase (EIF2AK) family, already implicated in NDD[42]. Moreover, GCN2 regulates neurogenesis, inhibits neuritogenesis, and influences behavioral control and memory[43], while PKR is involved in neurodegenerative diseases, including Alzheimer's[44] and dysregulated type I IFN-signaling[11,21]. This study identifies GCN2 as an additional player in ISR-mediated IFN-induction and highlights potential pharmacological interventions using PKR, GCN2, and JAK inhibitors (Fig. 8d and Supplementary Fig. 10).

Another remarkable pathophysiological consequence of *PSMC5* variants is the disruption of lipid homeostasis in ectodermal (Fig. 9c, d) and T cells (Fig. 7 and Supplementary Fig. 7), which recall the lipodystrophy in PRAAS subjects[40]. Similar findings have been observed in neuropsychiatric and neurodegenerative diseases, where cholesterol imbalance can impair neuronal structure and function[45,46].

Cells harboring *PSMC5* variants exhibit characteristics reminiscent of senescent or neurodegenerative cells, including impaired mitophagy[47], type I IFN response[48], abnormal apolipoproteins and/or cholesterol levels[49,50], cognitive inflexibility[51,52], impaired neuritogenesis[53,54], and an imbalanced E/I ratio[55,56]. The most striking shared feature is however a decline in proteasomal activity, accompanied by the formation of aggresomes[32,57], which is negatively correlated with lifespan in human multipotent cells[58] and in *Drosophila*[59]. Notably, the phosphorylated form of Psmc5/Rpt6 has been proposed as a marker for monitoring proteasomal activity in the aging rat brain[36], and the decreased expression of *PSMC5* as a potential biomarker for the onset or development of Parkinson's disease, Lewy body dementia or Alzheimer's disease[60–62].

To conclude, the combined inputs of T cells, iPSC-derived cells, neuronal models, animal studies, and cognitive paradigms contribute to a multi-dimensional understanding of the pivotal role of PSMC5/

Rpt6 in neurodevelopment and neuronal function. Revealing major overlaps in the molecular pathogenesis of NDD stemming from *PSMC5*, *PSMC3*, *PSMD11* and *PSMD12* variants[10–12], these findings not only enhance our understanding of the mechanisms underlying NDPs, but also lay the foundation for improved diagnostic procedures and potential therapeutic strategies. While the primary focus will be on individuals with these specific proteasome-related disorders, the remarkable similarities observed between the cellular characteristics of individuals with *PSMC5* variants and those with neurodegenerative diseases suggest that these therapeutic strategies could eventually benefit a broader population.

## Methods
Details regarding the reagents, equipment, software, oligonucleotide sequences, animal strains, and cell lines used in the following methodological sections are provided in Supplementary Table 5a–e.

### Genetic studies and ethics statement
Forty-four affected individuals, along with their healthy parents where possible, were included in this study. Enrollment was conducted by diagnostic or research teams as listed in Supplementary Table 6a. Written informed consent was obtained from all participants or their legally authorized representatives. Each affected individual was clinically evaluated, and written consent for use with the GestaltMatcher tool was secured for subjects S1, S3, S10, S12, S13, S15, S16, S23, S34-38, S40, and S44. Genome or exome sequencing was performed on all affected individuals, typically using a trio-based approach. All data included in this study were collected and shared in compliance with institutional policies and applicable regulations. Centers enrolling patients in a research context obtained approval from their local ethics committees (Supplementary Table 6b). For selected centers contributing data from a single patient in a diagnostic setting, IRB approval was not required, as the data were fully anonymized prior to transfer and excluded any photographic or personally identifiable information.

### Three-dimensional (3D) structural analysis of PSMC5 variants
To assess the spatial distribution of amino acid substitutions within PSMC5/Rpt6, we examined two high-resolution cryo-electron microscopy (cryo-EM) structures of the human 26S proteasome: one in a substrate-engaged state (PDB ID: 6MSK) and the other in a substrate-free conformation (PDB ID: 7W37). Structural mapping and visualization of the variants were performed using UCSF ChimeraX.

### Facial image analysis
Facial similarity among 15 *PSMC5* subjects (S1, S3, S10, S12, S13, S15, S16, S23, S34–38, S43, S44) was assessed using the GestaltMatcher (GMDB) approach[17], with parental consent. Each image was encoded via model ensemble and test-time augmentation into twelve 512-dimensional vectors. Pairwise similarity was quantified by averaging 12 cosine distances; lower values indicate greater resemblance.

To evaluate intra-cohort similarity, we compared mean pairwise cosine distances among *PSMC5* subjects to two control distributions derived from 1,555 GMDB images (328 syndromes)[63]: (1) subjects sharing a syndrome, and (2) randomly selected individuals. For each syndrome S and cohort size n ($2 \leq n \leq |S|$), we sampled 100 cohorts: $C_S$ (n patients from S) and $C_R$ (n random GMDB patients), excluding duplicates. Mean pairwise distances were computed for each.

Discriminative power was assessed via Receiver Operating Characteristic (ROC) analysis using mean pairwise distance as the metric, with 5-fold cross-validation. The optimal threshold ($c = 0.915$) was selected via Youden index, yielding sensitivity = 0.851, specificity = 0.862, and AUC = 0.895.

To test *PSMC5* cohort similarity, we sampled sub-cohorts $C_C$ of size $n$ ($2 \leq n \leq |C|$), computed their mean pairwise distances, and

repeated this 10,000 times (duplicates removed) to generate a reference distribution.

Finally, pairwise comparisons were performed across the 15 *PSMC5* subjects. Each was tested via leave-one-out cross-validation against 7459 GMDB images (449 disorders), ranking the remaining 14 *PSMC5* subjects within the broader image space.

## Behavioral studies in fly
**Background.** *Rpt6*, the *Drosophila* ortholog of *PSMC5*, shows strong homology (DRSC integrative ortholog prediction tool score 15/16). KD was performed using the GAL4-UAS system, based on truncating patient variants (V5:p.(Lys196Argfs29), V23:p.(Asp394Glufs2); Fig. 1a) predicted to impair proteasome function.

**Drosophila strains and crosses.** UAS-Rpt6 RNAi, ELAV-GAL4, and wild-type strains were used. Flies were reared on cornmeal agar at 22 °C and 40% humidity under University of Alberta biosafety regulations. Virgin ELAV-GAL4 or wild-type females were crossed to UAS-RNAi males.

**Olfactory learning.** Groups of 100 flies were trained to associate one odor with footshock[64], while a second odor was presented without. Learning was assessed by odor avoidance. Reciprocal experiments on 100 naïve flies controlled for odor bias. Performance index was calculated four times from both trials ($n = 4$ performance indexes per biological replicates of 200 flies).

**Reversal learning.** Flies received an additional training cycle with inverted odor–shock pairings prior to testing, requiring them to unlearn the initial association and acquire a new one. Reversed pairings were used on 100 naive flies to control for bias. Performance index was calculated from both trials ($n = 4$ biological replicates).

Assays were conducted at 25 °C, 70% humidity. Adults were transferred to bottles the day before testing and assayed at 1–3 days post-eclosion. Aged flies were maintained on standard food and transferred weekly to avoid contamination with next generation.

**Climbing[18].** Groups of 20 one-day-old flies were placed in vials the day before testing. During trials, flies were tapped to the bottom of a 250 ml glass cylinder, and the number reaching 17.5 cm within 2 min was recorded. Results were expressed as the percentage above the line at 10 s intervals ($n = 10$ biological replicates).

**Statistics.** Two-tailed *t*-tests were used for olfactory and climbing assays. Reversal learning was analyzed via one-way ANOVA followed by Tukey's post hoc test. All analyses were performed using Prism.

## *Psmc5* KD in rat hippocampal neurons
**Primary neuron culture.** Rat hippocampal neurons were seeded on poly-L-lysine (0.1 mg/mL)-coated coverslips in 24-well plates ($5 \times 10^4$ cells/well) and cultured for 14 days in Neurobasal medium supplemented with 2% B27, 1% Glutamax, and 100 μg/mL penicillin/streptomycin at 37 °C, 5% $CO_2$.

**Lentiviral KD.** Twenty-four hours post-seeding, *Psmc5* shRNA lentiviral particles were added at a multiplicity of infection of 5 for 18 h before medium replacement.

**Immunocytochemistry.** Neurons were fixed (4% PFA, 15 min), permeabilized (0.2% Triton X-100, 5 min), and blocked (10% FBS in PBS, 1 h). Primary antibodies—MAP2 (1:1000), vGlut (1:1000), vGAT (1:200), ubiquitin (1:500), and PSMC5 (1:400) in blocking solution—were applied overnight at 4 °C. After PBS washes (1X, $3 \times 5$ min), secondary antibodies (Alexa Fluor® 568) were incubated for 1 h in the dark. Cells

were washed (1X PBS, $2 \times 5$ min), then nuclei were counterstained with DAPI (300 nM, 5 min), followed by final PBS and ddH$_2$O washes.

Images were acquired at 40x magnification using an ImageXpress Micro Confocal microscope and analyzed with ImageJ v1.53. Each "n" represents the number of images analyzed per condition in each experiment.

## Neuronal morphology analysis in mouse
**Cloning and constructs.** Human *PSMC5* WT (NM_002805.6) and mutant cDNA sequences were PCR-amplified from pcDNA3.1/Zeo(+) expression vector, tagged with Asc/PacI sites, and ligated into a dual-promoter expression vector (CAGG-driven *PSMC5* expression; PGK-driven dtTomato for transfection tracking). An empty vector (EV) lacking *PSMC5* insert served as a control. The following human *PSMC5* variants were introduced in mutant constructs: c.601C>T p.(Arg201Trp), c.605C>T p.(Ala202Val), c.620C>T p.(Thr207Met), c.959C>A p.(Pro320His), c.959C>G p.(Pro320Arg), c.973C>T p.(Arg325Trp), and a deletion mimicking a splice site variant (Ala324Lys360del).

**Animals and ethics.** FvB/NHanHsd mice were group-housed and crossed under standard conditions with ad libitum food and water. All experiments complied with EU Directive 2010/63/EU (license AVD101002017893) and were approved by the Erasmus MC IRB.

**Primary hippocampal cultures.** Hippocampi from E16.5 embryos were dissected in ice-cold Neurobasal medium (NB), digested with Trypsin/EDTA (20 min, 37 °C), and washed in warm NB. Cells were dissociated in supplemented NB (1% penicillin/streptomycin, 1% GlutaMax, 2% B27), seeded on poly-D-lysine-coated coverslips in 12-well plates (1 mL/well), and incubated at 37 °C, 5% $CO_2$.

**Transfection and immunostaining.** Neurons were transfected on DIV3 with 2.5 μg DNA per *PSMC5* WT or mutant construct or 1.8 μg EV using Lipofectamine2000. Cells were fixed at DIV8 (4% PFA/sucrose), labeled with MAP2 antibody (1:500) and visualized with Alexa647-conjugated secondary antibody (1:200), and mounted in Mowiol for confocal imaging.

**Imaging and analysis.** Images were acquired using a Zeiss LSM700 confocal microscope (20× objective, 0.5 zoom, 2048 × 2048) from 10 neurons per condition per batch. Neurite length and branching were quantified using ImageJ with NeuronJ plugin. Data were normalized to *PSMC5* WT neurons and pooled across batches. Statistical analysis was performed using one-way ANOVA with Dunnett's post hoc test (GraphPad Prism). Each "n" represents a single traced neuron; ≥8 neurons per condition from ≥2 independent batches.

## T cell expansion from PBMC
Peripheral blood was collected from healthy donors and Subjects S1/7/14/15/25/36/37, along with available relatives. PBMCs were isolated via density gradient centrifugation and cryopreserved in 90% FBS/10 % DMSO. For expansion, PBMCs were co-cultured with irradiated allogeneic PBMCs in 96-well plates, supplemented with IL-2 and PHA-L (as in ref. 11). Cells were maintained in RPMI1640 with 10 % human AB serum and 1% penicillin/streptomycin, and analyzed at rest after 3–4 weeks.

## SDS-PAGE and immunoblotting
Resting T cells and/or SH-SY5Y cells were lysed in RIPA buffer; protein concentration was determined via BCA assay. Lysates (20 μg) were resolved on 10–15% SDS-PAGE, and transferred to PVDF membranes (wet transfer, 200 V, 400 mA, 1 h, 4 °C). Membranes were blocked (ROTI®Block, 20 min), incubated overnight at 4 °C with primary

antibodies targeting HA, PSMC5/Rpt6, α-tubulin, α6, K48-linked ubiquitin, GAPDH, GRP94, ATF6, BNIP3L. Detection was performed using HRP-conjugated secondary antibodies and chemiluminescence.

## RNA isolation, reverse-transcription and PCR analysis

Total RNA was extracted from snap-frozen SH-SY5Y and T cells, and 500 ng was reverse-transcribed (M-MLV). Real-time PCR was performed in duplicates to quantify ISGs (*IFIT1*, *IFI27*, *IFI44*, *IFI44L*, *ISG15*, *MX1*, *RSAD2*) and *GAPDH* using FAM-tagged TaqMan™ assays. Relative expression was calculated via the $2^{-\Delta\Delta Ct}$ method. IFN scores were derived as the median RQ of the seven ISGs relative to a calibrator (as in ref. [11]). Semi-quantitative PCR on SH-SY5Y cDNA amplified HA-*PSMC5* using primers targeting *PSMC5* and the polyadenylation signal of the pcDNA3.1/Zeo(+) vector. Products were resolved on 1.7% agarose gels, stained with GelRed®, and visualized under UV (312 nm).

## Native-PAGE and in-gel activity assays

Resting T cells from control and NDD patients with *PSMC5* variants underwent three freeze-thaw cycles (liquidN₂) in TSDG buffer. Supernatants were collected post-centrifugation (14,000 × *g*, 15 min, 4 °C) and protein quantified via Bradford assay. Extracts (20 µg) were resolved on 3–12% Bis-Tris gels; proteasome activity was assessed by incubating gels with 0.1 mM LLVY-AMC at 37 °C for 30 min. Gels were transferred to PVDF membranes and probed with antibodies against PSMC5/Rpt6 and α6. Detection was performed using HRP-labeled secondary antibodies and chemiluminescence.

## Proteostasis and mitophagy assays

T cells expanded from PBMCs of NDD subjects S7, S14, S15, S25 and S36 were stained with 1 µM PROTEOSTAT® dye and analyzed using the B3 (PerCP-Vio 700) channel (Supplementary Fig. 13).

In parallel, T cells expanded from subjects (*n* = 7; S1/7/14/15/25/36/37) and healthy donors (*n* = 7) were incubated overnight with 100 nM Mtphagy dye and analyzed by flow cytometry using the B4 (PE-Vio 770) channel.

## Lipid profile analysis

Lipid profiling was performed by high-resolution mass spectrometry coupled to reversed-phase UHPLC. Lipids were extracted from donor-derived PBMCs following a protocol slightly adapted from ref. [65]. Briefly, 225 µL cold MeOH (0.01 % BHT) was added to cell pellets, vortexed, pipettied, and sonicated (30 s, on ice). After addition of 3 µL EquiSPLASH™ and 750 µL cold MTBE, samples were incubated 1 h at 4 °C (650 rpm). Following addition of 188 µL ultrapure water and centrifugation (10,000 × *g*, 10 min), the upper organic phase (700 µL) was collected. The lower phase was re-extracted with 400 µL MTBE and 10 µL acetic acid (30 min, 4 °C), and 400 µL of the organic layer was pooled. After drying over sodium sulfate, solvents were evaporated under nitrogen and samples stored at –80 °C.

Dried extracts were rehydrated in chloroform/methanol/isopropanol (1:2:4, buffered with ammonium acetate pH8) and analyzed in duplicate using a Vanquish UHPLC coupled via nano-electrospray to a QExactive Plus mass spectrometer (positive/negative mode, four injections/sample). Separation was achieved at 50 °C using a linear gradient from buffer A (60:40 water:acetonitrile, 10 mM ammonium formate, 0.1% formic acid) to buffer B (90:10 isopropanol:acetonitrile, same additives), ramping from 20% to 99% B over 29 min. Spectra were acquired in DDA mode (loop count 15) with 70,000 resolution (MS1) and 17,500 (MS2); collision energy was set to 25. Spray voltage (HESI II) was 2.80 kV with polarity-specific adjustments.

Raw data were processed using LipidSearch software with precursor/product tolerances of 5/7ppm, m-score ≥2.0, and filters for top rank, main isomer peak, fatty acid priority, and ID quality A–B (Identification is limited to the full structure, e.g., head group, glycerol backbone, fatty acids). All lipid classes were included. Ion adducts: +H,

+NH₄, +2H (positive); –H, +HCOO⁻, –2H (negative). Identified species were filtered by peak quality (>0.8) and mass deviation (–1 to +1 ppm). Data were subsequently processed parsed in R (tidyverse); intensities were normalized to EquiSPLASH internal standards, technical replicates averaged, and samples median–median normalized with half-minimum imputation. Principal component analysis (PCA) was performed using the R-packages FactoMineR and Factoextra. Lipid class fold changes were calculated as ratios of mean lipid ion intensities of the patient group compared to the healthy donor group. Statistics were calculated using Student's *t*-test.

## NanoString profiling

Gene expression was assessed from 100 ng RNA extracted from control and affected T cells using the NanoString nCounter® Human AutoImmune Profiling Panel, following manufacturer's instructions. Data were normalized to housekeeping genes accordingly.

## iPSC generation and differentiation

*Generation of variant iPSC lines.* To assess the impact of the recurrent *PSMC5* variant c.973C>T p.(Arg325Trp), we introduced it by genome editing in the commercial iPSC line ASE-9211, devoid of any variants in UPS genes. CRISPR/Cas9 editing was conducted via nucleofection (4D-Nucleofector™, Lonza) of 1 × 10⁶ iPSCs with 200 pmol HDR template and a ribonucleoprotein complex (225 pmol crRNA, 225 pmol tracrRNA-ATTO+, 120 pmol Cas9; IDT). After 24 h, ATTO+ cells were sorted by FACS (MoFlo Astrios, CYTO-ICAN, Paris) and plated at low density (40–50 cells/cm²) on Synthemax II substrate with CloneR™2 supplement for clonal selection. Clones were picked after 7 days, expanded on Laminin-521, cryopreserved, and genotyped by PCR/Sanger sequencing to confirm variant incorporation. Chromosomal integrity was assessed using the iCS-digital™ PSC kit.

**iPSC culture and trilineage differentiation.** All procedures involving genetically modified organisms were approved by the French Ministry of Higher Education and Research (Article L.532-3). iPSCs were cultured in mTeSR™1 medium on Matrigel® hESC-qualified matrix. Trilineage differentiation was performed using ASE-9211, two edited WT clones (C10, E5), and three variant clones (G1, G8, H1) with the StemMACS™ Trilineage Kit. After 7 days, RNA was extracted, and 1 µg was reverse-transcribed. qPCR was performed using TaqMan assays for *SOX1*, *PAX6*, *HAND1*, *TBXT*, *FOXA2*, and *SOX17*.

**NPC differentiation.** NPCs were generated over 28 days using the STEMdiff™ Neural System from ASE-9211, three WT clones (B10, C10, E5), and three variant clones (G1, G8, H1). On day28, cells were seeded on coverslips (5 × 10⁵ cells/well), fixed (4% PFA), permeabilized (0.5% Triton X-100), and blocked (0.5% BSA). Immunostaining was performed with anti-SOX2 and anti-Nestin antibodies, followed by Alexa Fluor® 488/568-conjugated secondary antibodies (1:1,000). Nuclei were counterstained with DAPI (300 nM), and cells were imaged at 10× magnification (Nikon Eclipse Ti2). SOX2/Nestin-positive cell frequencies were quantified using ImageJ.

## Transcriptomics

**Data generation.** Two cell types were analyzed: (i) T cells from 12 healthy donors and four affected individuals (S15, S25, S36, S37); after culture, 1 × 10⁶ cells were lysed in 700 µL Qiazol and RNA extracted using the miRNeasy kit. (ii) Ectodermal cells derived from six iPSC lines: three with p.(Arg325Trp), two isogenic WT controls, and the commercial line ASE-9211, differentiated using the trilineage kit.

RNA quantity and integrity were assessed via Qubit-Flex fluorometer and LabChip GX Touch bioanalyzer. Libraries were prepared from 10 ng RNA using 3′ digital gene expression profiling (as in ref. [66]), incorporating universal adapters, barcodes, and unique molecular identifiers (UMIs) during template-switching reverse transcription.

Barcoded cDNAs were pooled, amplified, and fragmented to enrich 3′ ends. Libraries (350–800 bp) were sequenced on a NovaSeq 6000.

**Bioinformatics.** Primary analysis is detailed in Supplementary Information. Differential expression was performed using DESeq2[68] with pre-filtering (counts <10 removed), reducing genes from 28,307 to 15,990. Wald test identified differentially expressed genes (adjusted $p \leq 0.05$, Benjamini-Hochberg correction, $\log_2 FC \geq 0$). Pathway enrichment was conducted using clusterProfiler; figures were generated with ggplot2 in R.

## Proteomics of T cells (second set of analyses)

**20S proteasome purification.** T cells from 13 individuals ($n = 13$; four patient samples, including three separate samples for S14, S15, S25, plus a pooled sample for S36 and S37; nine healthy donors) were expanded from PBMCs ($80–100 \times 10^6$ cells/sample), lysed in HEPES buffer (10 mM HEPES pH 7.9, 1% NP40, 10 mM KCl, 10 mM EDTA, 10 mM ATP, 5 mM MgCl$_2$, protease/phosphatase inhibitors), sonicated (15 cycles, 30 s on/off, Bioruptor Pico), and centrifuged ($14,000 \times g$, 30 min, 4 °C). Supernatants (1–5 mg protein) were incubated overnight with MCP21 antibody (50 μg, anti-α2 subunit) cross-linked to Protein G MagBeads (50 μl 25% slurry) using 20 mM dimethyl pimelimidate in 0.2 M triethanolamine (pH 8.2). Beads were washed (Tris buffer with ATP/MgCl$_2$) and eluted in 5% SDS/50 mM ammonium bicarbonate (50 μL). Eluates were snap-frozen and stored at −70 °C for LC-MS/MS. Aliquots (20 μL) were retained for global proteome analysis.

**Bottom-up LC-MS/MS and data processing.** Lysates/eluates from protein samples of 13 individuals ($n = 11$; 5 patients; 6 healthy controls) were reduced (100 mM TCEP), alkylated (400 mM CA, 9 5 °C, 5 min), digested with trypsin (2% protein weight, ≥1 μg) using S-Trap™ micro spin columns, and desalted using Evotip. Peptides (2 μg) were analyzed by nanoLC (UltiMate 3000 RS, Thermo) coupled to TimsTOF SCP MS (Bruker) on a C18 Aurora column (25 cm × 75 μm, IonOpticks). Gradient: 2–20% B (30 min), 20–37% (3 min), 37–85% (2 min); flow rate: 150 nL/min. DIA-PASEF acquisition covered 400–1,000 m/z and 1/K$_0$ 0.64–1.37, with 8 TIMS ramps (100 ms accumulation, 3 MS/MS windows of 25Th). Collision energy ranged from 59 eV (1/K$_0$ = 1.6) to 20 eV (1/K$_0$ = 0.6).

Raw data were processed with DIA-NN1.9 using a predicted UniProt human library including PSMC5 variants and contaminants[67]. Results were validated in Proline[68] (FDR ≤ 1%, peptide length 7–30, charge 2–3). Significant peptides/proteins were identified using $t$-tests ($p < 0.01$), median ratio fitting, and normalization. Missing values (≤2) were imputed using the 5% centile; $t$-tests and $z$-tests were applied.

**Global proteome analysis.** Contaminant libraries were used to exclude background signals[67]. Protein abundances were normalized globally. Differential expression was visualized via volcano plots ($p < 0.05$, $\log_2 FC \geq \pm 1$) using ggplot2. Mitochondrial proteins were annotated using MitoCarta3.0 inventory[69].

**Proteasome interactome analysis.** IBAQ values were calculated by dividing protein abundance by observable peptides, normalized to total 20S content (average IBAQ of PSMA1-7, PSMB1-4). Stoichiometry and relative abundance of 20S subunits, activators, and interactors were computed as per Fabre et al.[70]. Comparisons used RM one-way ANOVA with Geisser–Greenhouse correction, followed by Šidák's multiple comparisons test with individual variance estimates.

## Statistics and reproducibility

No statistical method was used to predetermine sample size. No data were excluded from the analyses, except for proteomics (second set of analyses) outliers with a high proportion of missing values. The experiments were not randomized. The Investigators were not blinded to allocation during experiments and outcome assessment.

### Reporting summary

Further information on research design is available in the Nature Portfolio Reporting Summary linked to this article.

## Data availability

All data supporting the findings of this study are available within the Article and its Supplementary Information files. Source data underlying the figures are provided in the accompanying Source Data file. The Source Data file includes raw measurements, quantified values, and uncropped scans of gels and blots used to generate the main figures and Supplementary Figs. Mass spectrometry proteomics data have been deposited to the via in the PRIDE repository through the ProteomeXchange Consortium under accession codes PXD048558 and PXD058728 (https://www.ebi.ac.uk/pride/archive/projects/PXD048558; https://www.ebi.ac.uk/pride/archive/projects/PXD058728). Lipidomics data have deposited in a repository of the Metabolomics Workbench with the dataset identifier https://doi.org/10.21228/M8TK06. Transcriptomics and Nanostring data have been deposited in the Gene Expression Omnibus (GEO) under respective accession codes GSE288665 and GSE306813 (https://www.ncbi.nlm.nih.gov/geo/query/acc.cgi?acc=GSE288665; https://www.ncbi.nlm.nih.gov/geo/query/acc.cgi?acc=GSE306813). Source data are provided with this paper.

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

## Acknowledgements

For this work, S.K. was awarded research grants from the Agence Nationale de la Recherche (ANR) for the project ANR-21-CE17-0005 (UPS-NDDecipher) and as a partner of the European Joint Programme on Rare Diseases (EJP RD) for the project ANR-22-RAR4-0001-01 (UPS-NDDiag), from la Région des Pays de la Loire as part of the National Trajectory program for the project TN_2021_AAP_UPS-NDDECIPHER_-INSERM_154550, and from Nantes University Hospital (AOI 2021; grant RC22_0020) for the BioTND-UPS biobank. The EJP RD initiative received funding from the European Union's Horizon 2020 research and innovation program under grant agreement N°825575. E.K. received support from the German Research Foundation (RTG PRO 2719; SPP 2453 KR1915/11-1) and from COST (European Cooperation in Science and Technology) Action ProteoCure CA20113. F.E. is a recipient of an I-SITE NExT Junior Talent Chair. S Bézieau received financial support from the University Hospital Center (CHU) of Nantes for the BioTND-UPS biobank (PROG/09/72-03). The TND-UPS project led by S.B., S.K., and F.E. is funded for 3 years through the patronage of the Mutuelles AXA as part of its health program dedicated to supporting innovative research projects in France. Additional funding was provided by the National Institutes of Health (NIH) grant R01MH101221 to E.E.E., investigator at the Howard Hughes Medical Institute; the National Natural Science Foundation of China (grant 82201314) and the Fundamental Research Funds for the Central Universities starting fund (grant BMU2022RCZX038) to T.W.; the National Institutes of Health and the National Institute of Neurological Disorders and Stroke (NIH-NINDS grant K23NS119666) to S.S.; the European Reference Network (ERN) on Rare Congenital Malformations and Rare Intellectual Disability (ERN-ITHACA; EU Framework Partnership Agreement ID: 3HP-HP-FPA ERN-01-2016/739516) to A.M.C.G.; the ERN on Neurological Diseases ERN-RND to D.G.A. Part of this work was carried out on ICV-iPS, the iPS cell culture facility of ICM-Paris Brain Institute, funded by the "Investissements d'avenir" program (ANR-10- IAIHU-06) and is part of the DIM C-BRAINS, funded by the Conseil Régional d'Ile-de-France. KW received funding from German Federal Ministry of Research and Education (BMBF) under the Center of Excellence Initiative (grant number 03Z22DN12). M.H. and Z.S. were supported by the Czech Ministry of Health (grant number NU22-07-00165). This study was conducted as part of GEM-EXCELL, a network of excellence in genetics and genomics integrated within the French network of University Hospitals HUGO ('Hôpitaux Universitaires du Grand Ouest'). It was carried out under the aegis of the FHU GenOMedS, with the support of the Health cooperation group of University Hospitals of the Great West (GCS HUGO) and the National Alliance for Life Sciences and Health (Aviesan). The Deciphering Developmental Disorders (DDD) study represents independent research commissioned by the Health Innovation Challenge Fund (grant number HICF-1009-003). The authors warmly thank Robert Beyer and Anne Brandenburg for their excellent technical assistance. I.D.B., M.D.D., and M.K.L. are contributors to the Care for Rare Solve study, led by the Care for Rare Canada Consortium, Children's Hospital of Eastern Ontario Research Institute, Ottawa, ON, Canada. The authors are deeply grateful to Tayyaba Khan and Cherith Somerville for their efforts in clinical data collection and management. We acknowledge the biological resource center for biobanking (CHU Nantes, Hôtel Dieu, Centre de ressources biologiques (CRB), Nantes, F-44093, France [BRIF: BB-0033-00040]).

## Author contributions

S.K., S. Bézieau, F.E., and E.K. conceived the study and designed experiments, acquired funding, collected clinical data and biological samples, interpreted data, wrote and revised the manuscript; A.B.R., M.O., M.S., J.S., L.M.B., U.V., E.H., K.W., F.R.-D., J. Marcoux, M.P.B., A Droit and J. Poschmann conducted and/or designed omics analyses; T.W., W.D., S.C., S Marsac, I.M.W., A.T., C.F., D.O., L.J., J.A.R., A.Z., C.H., D Bonneau, E.T., A.B., K.G.M., S.V.M., C.M.L.V.T., K.L.I.v.G., R.O., M.S.d.P., K.S., A.R., I.I., K.M.D., E.B., A Dauber, J.B., N.A.V.F., R.A.B., T.N.T., S.S., K.A.D., L.C.S., C.C., A.A., R.K.J., J Pappas, R.R., D.N., A.C.H.T., K Kovak, D.B.B., M.C.V.M., D.R.A., L.W., R.D.G., C.C.M., D Babikyan, Z.S., M.H., A.T.T., H.A., B.N., K King, M J Hajianpour, G.C., D.P., C.L., D Geneviève, A Vitobello, A Sorlin, C.P., T.H., O.T., A Sabir, D.L., M J Hamilton, L.J.B., E.C., S Weber, T.L.H., A.M.C.G., E.F.T., D.G.A., M Codina-Solà, A Ververi, E.P., A.L., K.G., M.R., J.L., S.J.L., A.M.M., M.K.L., M.D.D., I.D.B., M.W., R.A.J., S.B.K., V.B., L Potocki, E.O., Y.S., E.W., M.L.T., M C.Schroeder, C.G., R.A.S., A.P., B.C., B.I., J Meiler, E.E.E. and K.M.W. contributed to the collection of genetic and clinical data by clinical assessment of patients and/or analysis of exome or genome sequencing data; C Rosenfelt and F.V.B. performed and/or designed experiments on *Drosophila melanogaster* and analyzed data; T.C.H. and P.M.K.performed and/or designed facial image analyses; J.E.S., G.M.v.W., L.B., J.J.P., C.d.K., V.V., M.S.T., T.B., A.M.H., F.G.T., S Wolfgramm, L.F., Y.V., A Dangoumau, L Poirier,

R.G., F.L., Y.E. and A.M.G. performed and/or designed in vitro cell model experiments and analyzed the data; C Ripoll and S Bigou performed genome editing of iPSC lines; V.M. and P.W.H. performed structural modeling. All authors reviewed the manuscript and approved its final version.

## Funding

## Competing interests
The Department of Molecular and Human Genetics at Baylor College of Medicine receives revenue from clinical genetic testing completed at Baylor Genetics Laboratory. E.E.E. is a scientific advisory board (SAB) member of Variant Bio, Inc. I.M.W., A.T., C.F., E.T., A.B., K.G.M., S.V.M., and K.M.W. are current or former employees of and may own stock in GeneDx, LLC. The remaining authors declare no competing interests.

## Additional information

Sébastien Küry [1,2] ✉, Janelle E. Stanton [3,4,116], Geeske M. van Woerden [5,6,7,116], Amélie Bosc-Rosati [8,9,116], Tzung-Chien Hsieh [10,116], Lise Bray [2,116], Marielle Oloudé [11,116], Cory Rosenfelt [12,116], Marie Pier Scott-Boyer [13], Victoria Most [14], Tianyun Wang [15,16,17], Jonas J. Papendorf [18], Charlotte de Konink [6,7], Wallid Deb [1,2], Virginie Vignard [2], Maja Studencka-Turski [18], Thomas Besnard [1,2], Anna M. Hajdukowicz [18], Franziska G. Thiel [18], Sophie Wolfgramm [18], Laëtitia Florenceau [2], Silvestre Cuinat [1,2], Sylvain Marsac [1], Yann Verrès [2], Audrey Dangoumau [19], Léa Poirier [2], Ingrid M. Wentzensen [20], Annabelle Tuttle [20], Cara Forster [21], Johanna Striesow [22], Richard Golnik [23], Damara Ortiz [24], Laura Jenkins [24], Jill A. Rosenfeld [25,26], Alban Ziegler [27], Clara Houdayer [28], Dominique Bonneau [28,29], Erin Torti [20], Amber Begtrup [20], Kristin G. Monaghan [20], Sureni V. Mullegama [20], Catharina M. L. Nienke Volker-Touw [30], Koen L. I. van Gassen [30], Renske Oegema [30], Mirjam S. de Pagter [30], Katharina Steindl [31], Anita Rauch [31,32,33,34], Ivan Ivanovski [31], Kimberly McDonald [35], Emily Boothe [36], Andrew Dauber [37], Janice Baker [38], Noelle Andrea V. Fabie [38], Raphael A. Bernier [39], Tychele N. Turner [40], Siddharth Srivastava [41], Kira A. Dies [41], Lindsay C. Swanson [41], Carrie Costin [42], Alali Abdulrazak [43], Rebekah K. Jobling [44], John Pappas [45,46], Rachel Rabin [45], Dmitriy Niyazov [47], Anne Chun-Hui Tsai [48], Karen Kovak [49], David B. Beck [50,51], May Christine V. Malicdan [52,53], David R. Adams [53], Lynne Wolfe [53], Rebecca D. Ganetzky [54,55], Colleen C. Muraresku [54], Davit Babikyan [56,57], Zdeněk Sedláček [58], Miroslava Hančárová [58], Andrew T. Timberlake [59], Hind Al Saif [60,61], Berkley Nestler [60], Kayla King [60], MJ Hajianpour [62], Gregory Costain [44,63,64], D'Arcy Prendergast [44,65], Chumei Li [66], David Geneviève [67], Antonio Vitobello [68,69], Arthur Sorlin [68,69,70], Christophe Philippe [68,69], Tamar Harel [71,72], Ori Toker [73], Ataf Sabir [74,75], Derek Lim [74,75], Mark J. Hamilton [76], Lisa J. Bryson [76], Elaine Cleary [77,78,79], Sacha Weber [80], Trevor L. Hoffman [81], Anna M. Cueto-González [82,83], Eduardo F. Tizzano [82,83], David Gómez-Andrés [84], Marta Codina-Solà [82,83], Athina Ververi [85], Efterpi Pavlidou [86], Alexandros Lambropoulos [87], Kyriakos Garganis [88], Marlène Rio [89], Jonathan Levy [90,91], Sarah J. Langas [92,93], Anne M. McRae [92], Mathieu K. Lessard [94], Maria Daniela D'Agostino [94], Isabelle De Bie [94], Meret Wegler [95], Rami Abou Jamra [95], Susanne B. Kamphausen [96], Viktoria Bothe [95], Lorraine Potocki [25,97], Eric Olinger [98], Yves Sznajer [98], Elsa Wiame [98], Michelle L. Thompson [99], Molly C. Schroeder [99,100], Catherine Gooch [101], Raphael A. Smith [102], Arti Pandya [102], Larissa M. Busch [103], Uwe Völker [103], Elke Hammer [103], Kristian Wende [22], Benjamin Cogné [1,2], Bertrand Isidor [1,2], Jens Meiler [14,104], Clémentine Ripoll [105], Stéphanie Bigou [105], Frédéric Laumonnier [19,106], Peter W. Hildebrand [107,108,109], Evan E. Eichler [110,111], Kirsty McWalter [20], Peter M. Krawitz [10], Florence Roux-Dalvai [13], Ype Elgersma [5,6], Julien Marcoux [8,9], Marie-

Pierre Bousquet [8,9], Arnaud Droit [13,112], Jeremie Poschmann [11], Andreas M. Grabrucker [3,4,113], Francois V. Bolduc [12,114,115], Stéphane Bézieau [1,2,117] ✉, Frédéric Ebstein [2,117] ✉ & Elke Krüger [18,117] ✉

[1]Nantes Université, CHU Nantes, Service de Génétique Médicale, Nantes, France. [2]Nantes Université, CHU Nantes, CNRS, INSERM, l'institut du thorax, Nantes, France. [3]Bernal Institute, University of Limerick, Limerick, Ireland. [4]Department of Biological Sciences, University of Limerick, Limerick, Ireland. [5]Department of Clinical Genetics, Erasmus Medical Center, Rotterdam, The Netherlands. [6]ENCORE Center of Expertise for Neurodevelopmental Disorders, Erasmus Medical Center, Rotterdam, The Netherlands. [7]Department of Neuroscience, Erasmus Medical Center, Rotterdam, The Netherlands. [8]Institut de Pharmacologie et de Biologie Structurale (IPBS), Université de Toulouse (UT), Toulouse, France. [9]Infrastructure Nationale de Protéomique, ProFI, UAR, Toulouse, France. [10]Institute for Genomic Statistics and Bioinformatics, University Hospital Bonn, Rheinische Friedrich-Wilhelms-Universität Bonn, Bonn, Germany. [11]Nantes Université, CHU Nantes, INSERM, Center for Research in Transplantation and Translational Immunology, UMR 1064, Nantes, France. [12]Department of Pediatrics, University of Alberta, Edmonton, AB, Canada. [13]Centre de recherche du CHU de Québec-Université Laval, Québec, QC, Canada. [14]Institute for Drug Discovery, Medical Faculty, Leipzig University, Leipzig, Germany. [15]Department of Medical Genetics, Center for Medical Genetics, Peking University Health Science Center, Beijing, China. [16]Neuroscience Research Institute, Peking University; Key Laboratory for Neuroscience, Ministry of Education of China & National Health Commission of China, Beijing, China. [17]Autism Research Center, Peking University Health Science Center, Beijing, China. [18]Universitätsmedizin Greifswald, Institut für Medizinische Biochemie und Molekularbiologie, Greifswald, Germany. [19]Université de Tours, INSERM, Imaging Brain & Neuropsychiatry iBraiN U1253, Tours, France. [20]GeneDx, LLC, Gaithersburg, MD, USA. [21]Loyola University Chicago, Chicago, IL, USA. [22]Leibniz Institute for Plasma Science and Technology (INP), Greifswald, Germany. [23]Department of Computer Science and Interdisciplinary Center for Bioinformatics, Bioinformatics Group, Universität Leipzig, Leipzig, Germany. [24]UPMC Children's Hospital of Pittsburgh, One Children's Hospital Drive, Pittsburgh, PA, USA. [25]Department of Molecular and Human Genetics, Baylor College of Medicine, Houston, TX, USA. [26]Baylor Genetics Laboratory, Houston, TX, USA. [27]Department of Medical Genetics, University Hospital of Toulouse, Toulouse, France. [28]Service de Génétique médicale, CHU Angers, Angers, France. [29]Mitovasc, UMR CNRS 6015, INSERM U1083, Angers University, Angers, France. [30]Department of Genetics, University Medical Centre Utrecht, Utrecht University, Utrecht, The Netherlands. [31]Institute of Medical Genetics, University of Zürich, Zurich, Switzerland. [32]University Children's Hospital Zurich, Zurich, Switzerland. [33]University of Zurich Research Priority Program ITINERARE: Innovative Therapies in Rare Diseases, Zurich, Switzerland. [34]University of Zurich Research Priority Program AdaBD: Adaptive Brain Circuits in Development and Learning, Zurich, Switzerland. [35]Norton Children's Medical Group, University of Louisville School of Medicine, Louisville, KY, USA. [36]University of Mississippi Medical Center, Jackson, MS, USA. [37]Division of Endocrinology, Children's National Hospital and Department of Pediatrics, The George Washington University School of Medicine and Health Sciences, Washington, DC, USA. [38]Department of Medical Genetics and Genomics, Children's Minnesota, Minneapolis, MN, USA. [39]Department of Psychiatry & Behavioral Sciences, Center on Human Development and Disability, University of Washington, Seattle, WA, USA. [40]Department of Genetics, Washington University School of Medicine, St. Louis, MO, USA. [41]Rosamund Stone Zander Translational Neuroscience Center, Department of Neurology, Boston Children's Hospital, Boston, MA, USA. [42]Department of Genetics, Akron Children's Hospital, One Perkins Square, Akron, OH, USA. [43]Division of Genetics, Department of Pediatrics, West Virginia University School of Medicine, One Medical Center Drive, Morgantown, WV, USA. [44]Division of Clinical and Metabolic Genetics, The Hospital for Sick Children, Toronto, ON, Canada. [45]Clinical Genetic Services, Department of Pediatrics, NYU Grossman School of Medicine, New York, NY, USA. [46]Clinical Genetics, NYU Orthopedic Hospital, New York, NY, USA. [47]Division of Medical Genetics, Department of Pediatrics, Duke University School of Medicine, Durham, NC, USA. [48]Department of Pediatrics, College of Medicine, University of Illinois, Chicago, IL, USA. [49]Department of Molecular and Medical Genetics, Oregon Health and Sciences University, OHSU, Portland, OR, USA. [50]Division of Rheumatology, Department of Medicine, New York University Grossman School of Medicine, New York, NY, USA. [51]Center for Human Genetics and Genomics, New York University Grossman School of Medicine, New York, NY, USA. [52]Medical Genetics Branch, National Human Genome Research Institute, NIH, Bethesda, MD, USA. [53]National Institutes of Health Undiagnosed Diseases Program, National Human Genome Research Institute, NIH, Bethesda, MD, USA. [54]Mitochondrial Medicine Program, Division of Human Genetics, Children's Hospital of Philadelphia, Philadelphia, PA, USA. [55]Department of Pediatrics, University of Pennsylvania, Perelman School of Medicine, Philadelphia, PA, USA. [56]Department of Medical Genetics, Yerevan State Medical University after Mkhitar Heratsi, Yerevan, Armenia. [57]Laboratory of Molecular Genetics, Center of Medical Genetics and Primary Health Care, Yerevan, Armenia. [58]Department of Biology and Medical Genetics, Charles University 2nd Faculty of Medicine and University Hospital Motol, Prague, Czech Republic. [59]Wyss Department of Plastic Surgery, NYU Langone Medical Center, New York, NY, USA. [60]Department of Human and Molecular Genetics, Division of Clinical Genetics, Virginia Commonwealth University School of Medicine, Richmond, VA, USA. [61]Department of Pediatrics, Division of Clinical Genetics, Virginia Commonwealth University School of Medicine, Richmond, VA, USA. [62]Division of Medical Genetics and Genomics, Department of Pediatrics, Albany Medical College, Albany, NY, USA. [63]Department of Molecular Genetics, University of Toronto, Toronto, ON, Canada. [64]Program in Genetics and Genome Biology, SickKids Research Institute, Toronto, ON, Canada. [65]Department of Paediatrics, Temerty Faculty of Medicine, University of Toronto, Toronto, ON, Canada. [66]McMaster University Medical Center, Hamilton, ON, Canada. [67]Université Montpellier, Inserm U 1183, Centre de référence maladies rares anomalies du développement, Service de génétique médicale, Hôpital Arnaud de Villeneuve, Montpellier, France. [68]UMR 1231 GAD, Inserm, Université de Bourgogne Franche Comté, Dijon, France. [69]Unité Fonctionnelle Innovation en Diagnostic Génomique des Maladies Rares, Fédération Hospitalo-Universitaire-TRANSLAD, CHU Dijon Bourgogne, Dijon, France. [70]Centre de Génétique et Centre de Référence Anomalies du Développement et Syndromes Malformatifs de l'interrégion Est et FHU TRANSLAD, Centre Hospitalier Universitaire de Dijon, Dijon, France. [71]Department of Genetics, Hadassah Medical Organization, Jerusalem, Israel. [72]Faculty of Medicine, Hebrew University of Jerusalem, Jerusalem, Israel. [73]Department of Pediatrics, Allergy and Clinical Immunology Unit, Shaare Zedek Medical Center, Faculty of Medicine, Hebrew University of Jerusalem, Jerusalem, Israel. [74]Clinical Genetics Department, Birmingham Women's and Children's NHS Foundation Trust, Birmingham, UK. [75]Institute of Cancer and Genomic Sciences, University of Birmingham, Birmingham, UK. [76]West of Scotland Clinical Genetics Service, Queen Elizabeth University Hospital, Glasgow, UK. [77]South East Scotland Genetics Service, Western General Hospital, Edinburgh, UK. [78]Centre for Clinical Brain Sciences, University of Edinburgh, Edinburgh, UK. [79]UK Dementia Research Institute at University of Edinburgh, University of Edinburgh, Edinburgh, UK. [80]Service de Génétique Médicale, Hôpital Armand-Trousseau, APHP, Sorbonne Université, Paris, France. [81]Department of Genetics, Southern California Kaiser Permanente Medical Group, Anaheim, CA, USA. [82]Department of Clinical and Molecular Genetics, Vall d'Hebron Hospital Universitari, Vall d'Hebron Barcelona Hospital Campus, Barcelona, Spain. [83]Medicine Genetics Group, Vall d'Hebron Institut de Recerca (VHIR), Vall d'Hebron Hospital Universitari, Vall d'Hebron Barcelona Hospital Campus, Barcelona, Spain. [84]Pediatric Neurology, Vall d'Hebron Institut de Recerca (VHIR), Vall d'Hebron Hospital Universitari, Vall d'Hebron Barcelona Hospital Campus, Barcelona, Spain. [85]Department of Genetics for Rare Diseases, 'Papageorgiou' General Hospital, Thessaloniki, Greece. [86]Department of Speech and Language Therapy, University Hospital of Ioannina, Ioannina, Greece. [87]Genetic Unit, 1st Department of Obstetrics and Gynecology, School of Medicine, Aristotle University of Thessaloniki, 'Papageorgiou' General Hospital, Thessaloniki, Greece. [88]Epilepsy Unit, St Luke's Hospital, Thessaloniki, Greece. [89]Service de Médecine

Génomique des Maladies Rares, Hôpital Necker-Enfants Malades, AP-HP, Paris, France. [90]Department of Genetics, APHP-Robert Debré University Hospital, Paris, France. [91]Multi-site medical biology laboratory SeqOIA—FMG2025, Paris, France. [92]Division of Genetics, Genomics and Metabolism, Ann & Robert H. Lurie Children's Hospital of Chicago, Chicago, IL, USA. [93]Department of Pediatrics, Northwestern University Feinberg School of Medicine, Chicago, IL, USA. [94]Division of Medical Genetics, Department of Specialised Medicine, McGill University Health Centre, Department of Human Genetics, McGill University, Montreal, QC, Canada. [95]Institute of Human Genetics, University of Leipzig Medical Center, Leipzig, Germany. [96]Institute of Human Genetics, University Hospital Magdeburg, University Hospital Magdeburg, Magdeburg, Germany. [97]Texas Children's Hospital, Houston, TX, USA. [98]Center for Human Genetics, Cliniques Universitaires Saint-Luc, UCLouvain, Brussels, Belgium. [99]Department of Pathology and Immunology, Division of Laboratory and Genomic Medicine, Washington University School of Medicine in Saint Louis, St. Louis, MO, USA. [100]Department of Pediatrics, Washington University School of Medicine, St. Louis, MO, USA. [101]Department of Pediatrics, Division of Genetics and Genomic Medicine, Washington University in St Louis, St. Louis, MO, USA. [102]Department of Pediatrics, Division of Genetics and Metabolism, University of North Carolina Health, Chapel Hill, NC, USA. [103]Universitätsmedizin Greifswald, Interfakultäres Institut für Genetik und Funktionelle Genomforschung, Abteilung für Funktionelle Genomforschung, Greifswald, Germany. [104]Department of Chemistry, Department of Pharmacology, Center for Structural Biology, Institute of Chemical Biology, Center for Applied Artificial Intelligence in Protein Dynamics, Vanderbilt University, Nashville, TN, USA. [105]ICV-iPS core facility, Sorbonne Université, Institut du Cerveau—Paris Brain Institute—ICM, Inserm, CNRS, APHP, Hôpital de la Pitié Salpêtrière, Paris, France. [106]Service de Génétique, Centre Hospitalier Régional Universitaire, Tours, France. [107]Institut für Medizinische Physik und Biophysik, Universität Leipzig, Medizinische Fakultät, Leipzig, Germany. [108]Charité Universitätsmedizin Berlin, Corporate member of Freie Universität Berlin and Humboldt-Universität zu Berlin, Institute of Medical Physics and Biophysics, Berlin, Germany. [109]Berlin Institute of Health, Berlin, Germany. [110]Department of Genome Sciences, University of Washington School of Medicine, Seattle, WA, USA. [111]Howard Hughes Medical Institute, University of Washington, Seattle, WA, USA. [112]Département de médecine moléculaire, Faculté de médecine, Université Laval, 2325 rue de l'Université, Québec, QC, Canada. [113]Health Research Institute (HRI), University of Limerick, Limerick, Ireland. [114]Neuroscience and Mental Health Institute, University of Alberta, Edmonton, AB, Canada. [115]Department of Medical Genetics, University of Alberta, Edmonton, AB, Canada. [116]These authors contributed equally: Janelle E. Stanton, Geeske M. van Woerden, Amélie Bosc-Rosati, Tzung-Chien Hsieh, Lise Bray, Marielle Oloudé, Cory Rosenfelt. [117]These authors jointly supervised this work; Stéphane Bézieau Frédéric Ebstein Elke Krüger. ✉e-mail: sebastien.kury@chu-nantes.fr; stephane.bezieau@chu-nantes.fr; frederic.ebstein@univ-nantes.fr; elke.krueger@uni-greifswald.de

