## [Transparent Peer review file · Nature Communications]

Investigating the neuronal role of the proteasomal ATPase subunit gene *PSMC5* in neurodevelopmental proteasomopathies

Corresponding Author: Dr Sébastien Küry

Version 0:

Reviewer comments:

Reviewer #1

(Remarks to the Author)

Review of paper by Kury et al. entitled Unveiling the crucial neuronal role of the proteasomal ATPase subunit gene *PSMC5* in neurodevelopmental proteasomopathies.

In this manuscript the authors identified 23 unique variants in *PSMC5* in 38 unrelated NDD individuals. The authors used a lot of different techniques and animal models (fly-mouse-rat) but to my opinion, these experiments were too superficial and not helping to elucidate the underlying general pathophysiology in the CNS.

The data obtained in the human T cells, on the other hand, is much more relevant and robust, but still do not fully explain the variety of clinical symptoms. I am missing the actual fundamental link between these mutations and the pathology.

Major remarks

Several mutations were “assumed de novo”. What does this mean exactly? Were parents tested yes or no? This is a very vague term to me.

Almost all patients carry a missense mutation. Patient S9 is homozygous for the mutation p.Thr207Met. This missense is also located in the AAA+ ATPase core domain, so one would expect that the heterozygous parents would also exhibit symptoms? Were they clinically examined, also for more subtle symptoms. It would also be interesting to take their T-cells along for the functional analyses.

Is there actually an average facial gestalt?

Given the dysregulation in type I IFN production, one would expect (auto)-immune problems. How thoroughly were the patients investigated for this? Any problems in the carrier parents of patients S5 and S17? They should be investigated.

The same holds true for the role of the proteasome in age-related neurodegenerative diseases. What is the age of the carrier mothers of patients S5 and S17? Were they investigated? Was segregation analysis done in these families? Maybe there are other family members that are older and carrier of the mutation.

Do proteasome-linked genes, and *PSMC5* in particular, pop-up in GWAS studies?

Minor remarks

L 268: delete second “in this context”

L 653: day one of development ? To what exact developmental time point does this correspond?

In the abstract, the authors state that they looked in “human” hippocampal neurons? Typo probably?

Reviewer #2

(Remarks to the Author)

Authors describe a total of 23 pathogenic *PSMC5* variants in 38 individuals of neurodevelopmental disorders with various other organs involvement. *PSMC5* encodes the AAA-ATPase proteasome subunit *PSMC5/Rpt6*, a component of the 26S proteasome complex. Several multimodal functional analyses have been performed: *PSMC5* overexpression resulting in altered human hippocampal neuronal morphology, *PSMC5* knockdown causing impaired reversal learning in *Drosophila* and loss of excitatory synapses in rat hippocampal neurons, abnormal proteasome assembly defects associated with most *PSMC5* variants, abnormalities of innate immune signaling, mitophagy and lipid metabolism together with their witness of PKR and GCN2 kinases potentially ameliorating immune dysregulations in affected patients. This reviewer thinks

identification of 23 novel variants and various functional assays are well-done, providing new scientific aspects in the new proteasomopathy, and suggest several points which may improve the manuscript. Recently, neurodevelopmental proteasomopathies with abnormalities of PSMD12, PSMC3, and PSMB1 have been reported, and PSMC5 abnormalities were added by this manuscript.

Majors

- Since the impact of each variant is complex and varies depending on the target of observation (e.g., effects on PSMC5 stability, proteasome assembly, mitophagy, IFN signature, etc), it is easier to understand if functional impacts by pathogenic variants are summarized in a figure or table.
- A scheme summarizing the causative genes and phenotypes of proteasome-related neurodevelopmental proteasomopathies and diseases with similar cellular dynamics to this disease, such as CANDLE and PRAAS, would be helpful for the understanding of readers.
- Are there any phenotype-genotype correlation among the position of variants regarding 3D structure, functional domains, in vitro phenotypes?
- Two affected mothers of S5 and S17 should be characterized at the similar level to their children. Perhaps include such detailed information in Tables 1, 2 and S2.
- Parents of S9 with a homozygous variant are assumed to have the variant heterozygously. Even if DNA is unavailable, their phenotypic information should be helpful to consider how the homozygous variant should be positioned in this manuscript. Please comment on the phenotypic difference between heterozygosity and homozygosity. Is this family consanguineous?
- Facial image analysis using GestaltMatcher is interesting. Please comment of the most typical face of PSMC5 variants if possible. S9 is a homozygous patient and others are all heterozygous patients, though.
- S26 with p.(Pro320His) has a twin with similar presentations and ADHD in the father. How about their genetic inheritance? The allele count for gnomAD also has 8 alleles.
- Different results between SHSY-5Y cell experiments and patients' T cells was considered to be due to compensatory effects. But this could be due to difference cell natures.
- Increased LMNA expression may cause premature aging. How about the accumulation of farnesylated pre-laminA (progerin)?
- Why do patients with PSMC5 variants show no obvious signs of autoinflammation?
- Are there any neurological symptoms that appear in late onset due to accumulation of ubiquitinated proteins in older patients?
- Since PSMC5 function may differ depending on the time of development and tissues, it is preferable to evaluate the pathophysiology using not only cultured cells but also knock-in mice.
- Are inclusion bodies composed of ubiquitinated abnormal proteins formed in neurons?

Minors

- What are three oval symbols covering three or four variants (above the protein) in Fig. S1?
- Line 639: two periods in the end of text.
- Line 689-697: No comments on p.(Gln221Arg).
- Line 708: p.(Thr210Met) should be p.(Thr207Met)?
- Line 705: p.(Arg201Trp) instead of p.(Arg201Val).
- Line 747: S1 is not analyzed in Fig. 6A.
- Line 790: wer should be were.
- Line 862: No D in Fig. S6.
- Line 1300-1316: Fig.7 B and C become D and E in legend.
- Figure 1: c.854G>A p.(Gly282Asp) should be c.854G>A p.(Gly282Asp).
- Only the first letter of the gene name of the model animals (fly and rat) are capitalized. All characters are capitals in Fig.3.
- Possible multigenic origin could be commented regarding S11 as the similar finding was reported in PMID: 37600812 (if applicable) as the same authors are involved.
- What are N50Rfs*7, L52P, M130I?, V362M and D394G in Figure S1?
- This reviewer does not understand what gnomAD (1) and gnomAD (2) in Table 1 mean. please fill in the allele frequency of gnomAD as Table S1.
- p.(Gly282Asp) is not shown in the 3D structural analysis figures.
- In Fig.7 B, S12's father is a typo of S11's father.
- In Fig.7 C, color-code or reshape the variants so that you can see the score of each variant. The right panel of Figure S5 B and the line graph below C should be similarly shown as indicated above.

Version 1:

Reviewer comments:

Reviewer #1

(Remarks to the Author)

Re-review of paper by Kury et al.

The authors have addressed my comments and also included additional data using a human iPSC model. These additional experiments have certainly improved the quality and content of the paper by providing more insight into the underlying pathophysiology.

Except for some minor typos, I have no additional questions.

Reviewer #2

(Remarks to the Author)

Authors added new experiments such as additional multi-omics analyses, iPSC model experiment, and recruited six new PSMC5-aberrant cases. Figures and Tables are newly structured. The iPSC experiment disclosed the impairment differentiation of neural progenitor cells, and additional multi-omics experiments showed abnormal mitochondrial integrity and protein quality control and dysregulated cholesterol metabolisms. Authors' efforts are significant. Though many data were statistically analyzed and sometimes showed subtle change, this reviewer recognizes authors are reasonably figure out neurodevelopmental proteosomopathies by PSMC5 aberration as shown below in response to this reviewer's previous concerns.

Majors

Remark 1: Table S1 is easy to understand the functional impact of variants.

Remark 2: Fig S1 effectively illustrates which and how variants of subunit genes of 26 proteasome lead to human disorders.

Remark 3: Very good to know that p.Arg325Trp and c.970-2A>G leading to exon 10 skipping containing Arg325 cause cardiovascular and musculoskeletal signs.

Remark 4: Two affected mothers are confirmed but unfortunately not described in detail because of their no-cooperation.

Remark 5: The complicated situation is understood by this reviewer.

Remark 6: This reviewer agrees that it is difficult to show an average portrait or a typical face.

Remark 7: Authors' consideration of p.(Pro320His) seems reasonable which can be described in the manuscript.

Remark 8: It is good to describe this reviewer's concern about different results came from different cells used.

Remark 9: This reviewer agrees with no mention of farnesylation of progerin as it was not reproducible.

Remark 10: As this reviewer expected, autoinflammation is likely involved and mentioned as the response to Remark 4 by the Reviewer 1.

Remark 11: Possible premature neurodegeneration is very interesting but exceptional and mentioned in the revised manuscript.

Remark 12: Instead of knock-in mouse, authors implemented an iPSC model to address c.973C>T p.(Arg325Trp).

Remark 13: OK.

Minors

Remark 16: Commented on p.(Gln221Arg).

Other minor remarks are well responded.

New other Errors and typos in the revised manuscript.

Page 6, line 247: Its. Knockdown>Its knockdown

Page 27, line 807: (7/44 subjects)>(6/44 subjects) and (8/44)>(9/44) based on the new Figure 1A

Figure 1B: Arg325Trp 8>9 and Pro320His 7>6 based on the new Figure 1A

Page 27, line 817-818: the American College of Medical Genetics and Genomics (ACMG) guidelines> the American College of Medical Genetics and Genomics/the Association for Molecular Pathology (ACMG/AMP) guidelines

Page 27, line 821: Fig. 1A>Fig. 2A

Page 32, line 952: (p.Arg325Trp)>p.(Arg325Trp)

Figure 5C: S23 (p.Pro320Arg)>S25 (p.Pro320Arg)

Figure 6A: S7 (V7)>S7 (V6)

Figure 7B: gene name is hard to see. This reviewer suggests a bit enlarged version is recommended.

Figure 7E legend: six NDD subjects, but this reviewer sees 7 red points in Figure 7E.

RESPONSE TO REFEREES

We would like to thank the referees for their insightful review of the manuscript and for their relevant comments, as well as yourself for processing this revision. In the amended version, we addressed all the comments made, including additional experiments and more patients to improve the quality of the manuscript and to meet the expectations of the reviewers.

To address the reviewers' remarks many new experiments in additional figures were required (new Figures: 2, 3D-J, 3M-N, 5C-D, 7A-B, 8C, 9, S1, S4, S5, S6A-B, S7D, S8, S11, S12) and led to substantial modifications of the text, figures and supplementary data. This includes additional multi-omics analyses and a human induced pluripotent stem cell (iPSC) model. In addition, six new cases were added compared to the previous version, which changed the numbering of affected subjects and variants. All redundancies across the tables and figures were removed, and we deleted Tables 1 and 2: (i) the variants are all reported in Fig. 1 and details regarding their nomenclature and pathogenicity are provided in Table S1 (in place of Table 1); whereas (ii) clinical findings are reported in Fig. 2 and detailed observations are listed in Table 2. We added a radar chart to indicate the main clinical features characterizing the disorder in Figure 2.

The new data indeed show impaired differentiation capacity of neural progenitor cells (NPGs) from ectodermal cells in our iPSC model, whereas differentiation into the 3 germ layers is less affected. These data explain a more central nervous system-directed manifestation of the disease. Additional multi-omics data indicate a critical involvement of PSMC5 in the maintenance of mitochondrial integrity and protein quality control, as well as support the dysregulated phospholipid metabolism also noted by lipidomics. Transcriptomics of iPSC-derived ectodermal cells revealed dysregulated differentiation signatures and altered cholesterol metabolism to support our major conclusions. Furthermore, these RNA sequencing experiments confirm a clear type I interferon signature in patient-derived T cells.

Our point-by-point responses to the comments made are listed below.

Reviewer #1 (Remarks to the Author):

General remark: Review of paper by Kury et al. entitled Unveiling the crucial neuronal role of the proteasomal ATPase subunit gene PSMC5 in neurodevelopmental proteasomopathies. In this manuscript the authors identified 23 unique variants in PSMC5 in 38 unrelated NDD individuals. The authors used a lot of different techniques and animal models (fly-mouse-rat) but to my opinion, these experiments were to superficial and not helping to elucidate the underlying general pathophysiology in the CNS.

The data obtained in the human T cells, on the other hand, is much more relevant and robust, but still do not fully explain the variety of clinical symptoms. I am missing the actual fundamental link between these mutations and the pathology.

- ✓ We thank the reviewer for the valuable comments and suggestions.

We agree that the clues provided in the previous version of the manuscript on the pathophysiological mechanisms leading from genotype to phenotype may have seemed insufficient. Consequently, we have designed additional experiments to provide further clues. On the one hand, (i) we have implemented an iPSC model. On the other hand, (ii) we have extended some of the cellular and animal models previously presented and (iii) included more multi-omics analyses.

- (i) iPSC model: We focused our assays on the c.973C>T p.(Arg325Trp) variant, as it is the most recurrent variant in the series, is unambiguously pathogenic according to functional explorations, and has a particularly marked multi-system action (cranio-facial system, heart, digestive system and ears) compared to the other variants (Fig. 2C). We assessed the pluripotency of the clones harboring this variant as well as their ability to differentiate into the three germ layers, including notably ectoderm, and into neural progenitor cells (NPC).

In brief, we found that the effects of the variant were initially well compensated by the iPSC, but became visible in later developmental stages. Thus, the pluripotency of the cell line was not significantly affected, as the iPSC could differentiate in the three germ layers (Fig. 9B; results: lines 1063-80), but the differentiation into neural progenitor cells was impaired (Fig. 9E-G; results: lines 1086-96). This problem of differentiation was already suggested by the under-expression of *HES1* as a target of Notch signaling observed by transcriptomics of ectodermal cells harboring the variant (Fig. 9C; results: lines 1080-2). *HES1* acts a transcriptional repressor and is known as a master regulator of neurogenesis. This observation alone establishes the genotype-phenotype correlation that was missing in the first version, showing that the developmental impact of the *PSMC5* variant occurs during embryogenesis, at the neuronal progenitor stage (discussion: lines 1096-9).

- (ii) Rat hippocampal neurons: Complementary analyses showed that *Psmc5* knockdown induced the aggregation of polyubiquitinated proteins, notably in the soma (Fig. 3M-N; results: lines 898-901). This finding should be considered in relation to the imbalance in the E/I ratio observed previously (discussion: lines 1127-38).

Drosophila: In addition to the impairment of reversal learning previously highlighted, climbing was also negatively and very significantly impacted by *rpt6* (*PSMC5*) knockdown (Fig. 3D-J; results: 883-9). Considering also the three possible cases of developmental regression (compared to milestones previously reached for language, cognition and or motor skills) noted in the cohort, this suggests that *PSMC5* variants might induce premature aging or neurodegeneration (discussion: lines 1148-50).

T cells from affected individuals: Transcriptomics and proteomics of more samples than in the previous version of the manuscript confirmed the dysregulation of type I interferon (IFN) signaling (Fig. 8C; results: lines 1035-6), and altered ribosome biogenesis (Fig S8A; results: lines 1007-11). Besides, the dysregulation of cholesterol metabolism highlighted by lipidomics on patient T cells was also suggested through deregulation of *MSMO1* by iPSC-derived ectodermal cells harboring variant c.973C>T p.(Arg325Trp) (Fig. 9C; results: lines 1082-4). Exploring these pathophysiological avenues could be the topic of future studies.

Major remarks

Remark 1: "Several mutations were "assumed de novo". What does this mean exactly? Were parents tested yes or no? This is a very vague term to me."

- ✓ The reviewer is right that the use of this term was not fully appropriate. It is officially used when a variant was found *de novo* but the biological relationship was not checked, which is automatically done when trio-exome or -genome sequencing is performed. We admit that we had misused the term here, by extending the notion to exceedingly rare variants not proven *de novo* by segregation analysis, but harbored by individuals whose biological parents were clinically unaffected. Consequently, in such situation, we corrected the mistake and simply stated that the variants were heterozygous and that parental samples were unavailable (Table S1).

Remark 2: "Almost all patients carry a missense mutation. Patient S9 is homozygous for the mutation p.Thr207Met. This missense is also located in the AAA+ ATPase core domain, so one would expect that the heterozygous parents would also exhibit symptoms? Were they clinically examined, also for more subtle symptoms. It would also be interesting to take their T-cells along for the functional analyses."

- ✓ We are aware that the presence of this unique homozygous variant could somehow be confusing. As a matter of fact, this observation does not challenge the mode of inheritance, which is autosomal dominant. Unfortunately, we are not allowed to disclose in the manuscript any information regarding the biological parents. In a confidential manner, we can inform the reviewers that the affected child (Subject 10 in the revised version of the manuscript) was born of incest between an uncle and his

niece, and is now under the authority of another family member. It is not unlikely that the two parents have NDD, but we cannot prove it.

A more convincing argument is the fact that we identified the same variant in an unrelated individual with NDD (Subject 11). Here again, segregation analysis could not be performed, and the clinical information provided is limited. Despite this lack of details, the report of this case shows that the variant in itself is associated with NDD at heterozygous state (see Table S2).

This is also in line with our functional studies showing that the variant p.Thr207Met had a negative effect on neuritogenesis.

Remark 3: Is there actually an average facial gestalt?

- ✓ We decided not to work on an average facial gestalt. Indeed, from our experience with other rare conditions, the series of individuals who provided photos is too small to generate significant results. The dysmorphic similarities between individuals are subtle and would be diluted in an average facial gestalt that would correspond to the face of a child with virtually no dysmorphic features. Therefore, we considered that it would be more informative to calculate the distance between the facial traits of the individuals compared to the large GestalMatcher database gathering thousands of photos from individuals with rare NDD.

Remark 4: Given the dysregulation in type I IFN production, one would expect (auto)-immune problems. How thoroughly were the patients investigated for this?

- ✓ The reviewer is right. Clearly, none of the patients has been primarily linked to auto-immunity syndromes such as proteasome associated autoinflammatory syndromes (PRAAS). Yet, patients have not been specifically assessed for inflammatory or immune problems, and such problems were probably not emphasized by the parents who were mainly concerned by developmental issues. However, we realized that our description was probably not accurate enough, as many of the observations reported in Table 2 as 'other clinical findings' are consistent with symptoms seen in PRAAS and other autoinflammatory syndromes. Among them are lymphadenopathy, periodic fever in Subject 4 (S4); skin anomalies in S6; erythema and bilateral vascular congestion of lower extremities in S9; recurrent ear infections, chronic cough and frequent respiratory illness in S12; diabetes with arterial hypertension in S14; recurrent infections in S17; recurrent chest infections, chronic cough in S18; humoral immunodeficiency in S20 and S21; bone marrow failure, first presenting with congenital thrombocytopenia in S35; congenital thrombocytopenia (megakaryocytic) in S37; congenital thrombocytopenia and anaemia in S38; iron deficiency anemia in S41; recurrent ear infections in S43.

In conclusion, there are links to autoimmunity from the clinical assessment of patients carrying *PSMC5* variants and there is overlap with PRAAS. We

therefore carefully discussed these overlaps in a relevant paragraph accordingly (lines 1165-73).

Remark 5: Any problems in the carrier parents of patients S5 and S17? They should be investigated. The same holds true for the role of the proteasome in age-related neurodegenerative diseases. What is the age of the carrier mothers of patients S5 and S17? Were they investigated? Was segregation analysis done in these families? Maybe there are other family members that are older and carrier of the mutation.

- ✓ Patient S5's mother (S6; aged 31 years), also harboring variant c.587del p.(Lys196Argfs*29), and patient S20 (formerly S17)'s mother (S21; aged 27 years) were enrolled in the study. Both were affected with symptoms similar to their children, confirming the pathogenicity of their respective variants, in line with the bioinformatics predictions and functional assessments. However, these two relatives are not old enough to provide any clues as to the possible involvement of proteasomal dysfunction in neurodegenerative diseases.

No further segregation analysis could unfortunately be performed in the two families, since because they did not respond to clinicians' invitations to investigate further.

- ✓ However, there is a body of evidence in favor of the involvement of proteasome dysfunction, related notably to *PSMC5*, in neurodegenerative diseases:
 - *PSMC5* deregulated expression would be a potential marker of the occurrence of Parkinson disease (PMID: 33042444)
 - *PSMC5* mRNA levels are significantly reduced in brain specimens from individuals with Parkinson's disease dementia, dementia with Lewy bodies or Alzheimer's disease (PMID: 28269775); these levels are associated with cognitive impairment in dementia with Lewy bodies.
 - *PSMC5* deregulation is included in a high risk AD signatures based on the APOE genotype (PMID: 20479757)
 - Decreased *PSMC5* mRNA levels could significantly predict the occurrence of Parkinson's disease (PMID 33042444).

Remark 6: Do proteasome-linked genes, and PSMC5 in particular, pop-up in GWAS studies?

- ✓ Numerous associations have been demonstrated between proteasomal gene variants and a wide variety of traits. If we limit ourselves to the genes we know best:
 - *PSMD12* variants have been associated essentially with body mass index (<https://www.ebi.ac.uk/gwas/genes/PSMD12>);
 - *PSMD11* variants have been associated with systolic blood pressure, but also to ADHD and schizophrenia (<https://www.ebi.ac.uk/gwas/genes/PSMD11>);
 - *PSMC3* variants emerged from more GWAS, including notable studies dealing with depressive symptoms, neurocognition,

cognitive function measurement, intelligence and Alzheimer (<https://www.ebi.ac.uk/gwas/genes/PSMC3>)

- *PSMC5* variant were highlighted in six studies, including one dealing with IGF-1 measurement and another one with memory decline in normal cognition

(<https://www.ebi.ac.uk/gwas/genes/PSMC5>). This last association (PMID: 37984853) is in line with our observations in *Drosophila* discussed in the manuscript.

- ✓ Beyond GWAS, the 26S proteasome and certain other proteasome genes have been identified as hub genes involved in early onset schizophrenia in a recent preprint (<https://doi.org/10.21203/rs.3.rs-5833160/v1>). Furthermore, a large exome-based international study on developmental disorders point to a significant enrichment of *de novo* rare *PSMC5* variants–predicted pathogenic–in the cohort (PMID: 33057194), which is consistent with our observations. This is indicated in the manuscript (discussion: lines 1095-8)

PSMC5 was also proposed as a prognostic biomarker for the survival to numerous cancerous disorders, including hepatocellular carcinoma survival (PMID: 38370535), lung adenocarcinoma (PMID: 35647031), glioma (PMID: 34804978) or breast-cancer (PMID: 34329194). *PSMC5* would also participate in the pathophysiological mechanism of COVID-19 and would thus represent a potential therapeutic target (PMID: 34004362).

Aside from these links with malignancies, studies of particular interest highlight the involvement of *PSMC5* in the pathogenesis of neurological diseases, as it would for instance drive the trinucleotide expansions involved in disorders such as Huntington’s disease (PMID: 23620289). Furthermore, its mRNA levels are decreased in blood samples from children with cerebral malaria (PMID: 28934429).

We did not mention the aforementioned articles in the manuscript, as they did not directly relate to the disorder we describe or to its clinical course. On the other hand, we cited three studies proposing *PSMC5* as potential biomarker (main text; lines 1186-8, see above). In more details:

- *PSMC5* deregulated expression would be a potential marker of the occurrence of Parkinson disease (PMID: 33042444)
- *PSMC5* mRNA levels are significantly reduced in brain specimens from individuals with Parkinson’s disease dementia, dementia with Lewy bodies or Alzheimer’s disease (PMID: 28269775); these levels are associated with cognitive impairment in dementia with Lewy bodies.
- *PSMC5* deregulation is included in a high risk AD signatures based on the APOE genotype (PMID: 20479757)

Remark 7-9: Minor remarks

- Remark 7: L 268: delete second “in this context”
 - ✓ It is done (lines 300-1).

- Remark 8: L 653: *day one of development? To what exact developmental time point does this correspond?*
 - ✓ We suppose that this term may have been confusing. Actually, it corresponds to day in vitro 1 (DIV1), which we used as a replacement of "day one of development" in the revised version of the manuscript (line 893).

Here, rat hippocampal neurons were purchased from Innoprot. They were isolated from embryonic day 18 rat brain hippocampus and were plated onto a substrate at DIV0, before lenti-virus infection at DIV1.

With the term "development" initially used, we intended to reflect early differentiation, comparable to the *in vivo* development of neurons transitioning from late embryonic to early postnatal stages.
- Remark 9: *In the abstract, the authors state that they looked in "human" hippocampal neurons? Typo probably?*
 - ✓ We apologize for the mistake. It was corrected to "mouse" (line 249).

Reviewer #2 (Remarks to the Author):

Authors describe a total of 23 pathogenic PSMC5 variants in 38 individuals of neurodevelopmental disorders with various other organs involvement. PSMC5 encodes the AAA-ATPase proteasome subunit PSMC5/Rpt6, a component of the 26S proteasome complex. Several multimodal functional analyses have been performed: PSMC5 overexpression resulting in altered human hippocampal neuronal morphology, PSMC5 knockdown causing impaired reversal learning in Drosophila and loss of excitatory synapses in rat hippocampal neurons, abnormal proteasome assembly defects associated with most PSMC5 variants, abnormalities of innate immune signaling, mitophagy and lipid metabolism together with their witness of PKR and GCN2 kinases potentially ameliorating immune dysregulations in affected patients. This reviewer thinks identification of 23 novel variants and various functional assays are well-done, providing new scientific aspects in the new proteasomopathy, and suggest several points which may improve the manuscript. Recently, neurodevelopmental proteasomopathies with abnormalities of PSMD12, PSMC3, and PSMB1 have been reported, and PSMC5 abnormalities were added by this manuscript.

We thank the reviewer for the valuable comments and suggestions.

Majors

Remark 1: Since the impact of each variant is complex and varies depending on the target of observation (e.g., effects on PSMC5 stability, proteasome assembly, mitophagy, IFN signature, etc), it is easier to understand if functional impacts by pathogenic variants are summarized in a figure or table.

- ✓ This is an excellent suggestion. The findings are now summarized in Table S1.

Remark 2: A scheme summarizing the causative genes and phenotypes of proteasome-related neurodevelopmental proteasomopathies and diseases with similar cellular dynamics to this disease, such as CANDLER and PRAAS, would be helpful for the understanding of readers.

- ✓ Thank you for the suggestion, all known proteasome gene variants and phenotypes summarized in the new Fig. S1. In addition, a review published recently summarizes all known proteasome gene variants and the clinical complexity of these proteasomopathies (PMID: 39220754). It is now cited in the manuscript (lines 291 and 1167).

Remark 3: Are there any phenotype-genotype correlation among the position of variants regarding 3D structure, functional domains, in vitro phenotypes?

- ✓ The proposed analysis is particularly relevant, considering the size of the case series presented. While preparing the earlier version of the manuscript, we had explored potential genotype-phenotype correlations but had not identified any significant findings. However, during the revision of Figure 1, some trends have emerged that are worth noting. It appears that

cardiovascular (conotruncal malformations) and musculoskeletal signs are more particularly prominent in association with the p.Arg325Trp variant and the splice variant c.970-2A>G predicted to induce skipping of the exon 10 coding the region containing the p.Arg325Trp variant. Besides, all the patients harboring this variant also have feeding difficulties, and several of them have hearing loss, two being sensorineural. By contrast, we noted that patients with variants affecting the first part of the protein frequently presented with short stature.

We mentioned these observations in the results section (results: lines 860-9).

Remark 4: Two affected mothers of S5 and S17 should be characterized at the similar level to their children. Perhaps include such detailed information in Tables 1, 2 and S2.

- ✓ The reviewer is right. Both were affected with symptoms similar to their children, confirming the pathogenicity of their respective variants, in line with the bioinformatics predictions and functional assessments. Please, see also our answer to reviewer 1's remark 5.

Remark 5: Parents of S9 with a homozygous variant are assumed to have the variant heterozygously. Even if DNA is unavailable, their phenotypic information should be helpful to consider how the homozygous variant should be positioned in this manuscript. Please comment on the phenotypic difference between heterozygosity and homozygosity. Is this family consanguineous?

- ✓ We fully agree and tried as much as possible to get information, but it was impossible. Unfortunately, we are not allowed to disclose in the manuscript any information regarding the biological parents. In a confidential manner, we can inform the reviewers that the affected child (now Subject 10 in the revised version of the manuscript) was born of incest between an uncle and his niece, and is now under the authority of another family member. It is not unlikely that the two parents have NDD, but we cannot prove it.

On the other hand, we included an additional patient heterozygous for the very same variant. Please, see our answer to reviewer 1's remark 2.

Remark 6: Facial image analysis using GestaltMatcher is interesting. Please comment of the most typical face of PSMC5 variants if possible. S9 is a homozygous patient and others are all heterozygous patients, though.

- ✓ Although most of the patients have dysmorphic facial features, these ones are very heterogeneous. It is therefore very difficult to draw a typical face of a patient affected by the PSMC5-related NDD. As discussed for reviewer 1's remark 3, we are not convinced by the interest of an average facial gestalt, considering the small number of patients contributing to facial analysis. The similarities between the patients clearly emerge when calculating the distance separating the patient with PSMC5 patients from

others, but this method takes into account subtle traits that would not appear on an average portrait.

Remark 7: S26 with p.(Pro320His) has a twin with similar presentations and ADHD in the father. How about their genetic inheritance? The allele count for gnomAD also has 8 alleles.

- ✓ We agree that this information would be invaluable. Unfortunately, the family did not reply to our request for a segregation analysis. While we will continue our efforts, it is unlikely that we will obtain any results of segregation analysis in time. On the other hand, bioinformatic predictions and molecular data strongly support the pathogenicity of this variant. Although its relative frequency does not provide a strong argument for pathogenicity, we have accounted for it in our predictions. Moreover, gnomAD v4 includes individuals who may have neurological or neuropsychiatric conditions (e.g., bipolar disorder, schizophrenia, depression), which could potentially be linked to the *PSMC5* variant. When excluding the UK Biobank data and analyzing gnomAD version v4.1.0 (non-UKB), only two individuals heterozygous for c.959C>A p.(Pro320His) are identified. Therefore, we do not consider this frequency to be a sufficient reason to dismiss the pathogenicity of the p.(Pro320His) variant. However, the relative frequency does suggest that its effects are likely less severe than those of the recurrent c.959C>G p.(Pro320Arg) variant, which affects the same nucleotide. This conclusion aligns with our clinical and functional observations.

Remark 8: Different results between SHSY-5Y cell experiments and patients' T cells was considered to be due to compensatory effects. But this could be due to difference cell natures.

- ✓ We thank the referee for raising this valid point and appreciate the opportunity to elaborate on this aspect. The contrasting protein levels of the p.(Arg201Trp), p.(Glu250Val), p.(Glu250Val), and p.(Arg325Trp) *PSMC5* variants observed between SH-SY5Y cells and T cells may have several explanations. As mentioned in the original manuscript, one possibility is that T cells exhibit unbalanced allele expression with the wild-type allele being expressed at higher levels than the mutant counterpart to compensate for *PSMC5* deficiency. In addition, as suggested by the referee, these differences could arise from the inherent characteristics of the two cell types or even from differences in experimental setups. Unlike T cells which endogenously express the variants, SH-SY5Y cells were transfected to ectopically express them from a plasmid-driven viral promoter at very high levels. Overexpression of individual proteasome subunits can disrupt the stoichiometric balance, rendering them more prone to degradation (PMID: 34382105). Thus, it is plausible that the SH-SY5Y system is more sensitive to detecting subtle effects of the variants on proteasome subunit turnover compared to T cells. This point has now been clarified and discussed in more detail in the revised version of the manuscript (results; lines 966-8).

Remark 9: Increased LMNA expression may cause premature aging. How about the accumulation of farnesylated pre-laminA (progerin)?

- ✓ Since the previous version of the manuscript, we performed additional proteomic analyses on a larger number of individuals than in the first series of proteomic analyses. The results obtained are therefore more robust. In this case, we were unable to reproduce the increase in LMNA levels as an event shared by all the patients tested. Consequently, we did not further investigate the possibility of farnesylated pre-lamin A accumulation.

Even if LMNA is eventually not deregulated in all patients, other findings in the second proteomics series suggest that the hypothesis of premature aging or neurodegeneration caused by *PSMC5* dysfunction remains worth exploring. In *PSMC5* patients, we noted for example increased expression levels of clusterin (CLU), which was proposed as a potential biomarker for Alzheimer's disease (PMID: 38017342). Yet, we are aware that such observations are fragile and would need to be replicated in more patients. For this reason, we have remained cautious in not mentioning them here, and instead focusing on the most significant results observed in the omics analyses.

Remark 10: Why do patients with PSMC5 variants show no obvious signs of autoinflammation?

- ✓ This comment is quite similar to reviewer 1's remark 4 about (auto)immune problems. We realized that our words may have sounded misleading. As a matter of fact, patients with *PSMC5* variants do not show the striking autoinflammatory symptoms that are the hallmark of PRAAS. However, when the clinical information provided by clinicians is dissected, many of the symptoms present in multiple patients in the series are suggestive of, or compatible with, immune problems (Table S1). These symptoms are detailed in the answer to reviewer 1's remark 4. Besides, we modified the related paragraph in the discussion (lines 1167-70).

A careful clinical re-assessment of all the patients focused on autoimmunity and immune dysregulation would be relevant, but not feasible in time. We are however convinced that the publication of this study will raise awareness that proteasomopathies are linked to type I interferon dysregulation and autoimmunity.

Remark 11: Are there any neurological symptoms that appear in late onset due to accumulation of ubiquitinated proteins in older patients?

- ✓ Very few adult patients are included in the cohort, and the observations are thus limited. However, brain MRI showed cerebral atrophy or a progression of brain lesions in three young adults, which could reflect neurodegeneration

and strengthen the importance of clinical follow-up to detect possible signs of premature neurodegeneration (discussion; lines: 1150-4).

Besides, in other neurodevelopmental proteasomopathies that we investigate in parallel, we are aware of cases of bipolar disorder or schizophrenia, hence our answer above regarding variant p.(Pro320His). We have not found yet any case of Alzheimer's or Parkinson's disease, but the few adults included are still relatively young. The cohort presented in this study on PSMC5 is therefore not necessarily the most suitable for studying this type of disease progression. Only the cases of regression, in particular those mentioned in the manuscript, support the idea of possible neurodegenerative mechanisms. Here again, further specific studies will be needed to validate our hypotheses.

Remark 12: Since PSMC5 function may differ depending on the time of development and tissues, it is preferable to evaluate the pathophysiology using not only cultured cells but also knock-in mice.

- ✓ We fully agree with the reviewer. However, there is no knock-in mouse model available, and it would have taken too much time to generate and implement it. To avoid revision delays, we implemented an iPSC model, focusing on the c.973C>T p.(Arg325Trp) variant, as explained at the beginning of the letter (see our answer to reviewer 1' general remark). We informed the editor about this choice of model.

Remark 13: Are inclusion bodies composed of ubiquitinated abnormal proteins formed in neurons?

- ✓ Yes, indeed. By immunostaining using anti-ubiquitin antibodies, we showed an accumulation of ubiquitinated proteins in *Psmc5*-KD hippocampal neurons (Fig. 3M,N; lines 898-901).

Minors

Remark 14: What are three oval symbols covering three or four variants (above the protein) in Fig. S1?

- ✓ We used these cloud-like symbols to highlight clusters of variants. We specified it in the caption of Fig. S2 (formerly Fig. S1).

Remark 15: Line 639: two periods in the end of text.

- ✓ It was deleted.

Remark 16: Line 689-697: No comments on p.(Gln221Arg).

- ✓ The reviewer is right. We indicated that the steady-state protein expression levels of PSMC5/Rpt6 were mildly reduced by variant p.(Gln221Arg). In

addition, we revised some of our interpretations for other variants compared to the previous version of the manuscript (results: lines 932-8).

Remark 17: Line 708: *p.(Thr210Met)* should be *p.(Thr207Met)*?

- ✓ The error of nomenclature is now corrected (results: line 952).

Remark 18: Line 705: *p.(Arg201Trp)* instead of *p.(Arg201Val)*.

- ✓ It is corrected (results: line 949).

Remark 19: Line 747: *S1* is not analyzed in Fig. 6A.

- ✓ We modified the text to distinguish between Fig. 6A and Fig. 6B -where S1 was included in the analyses (results: lines 989-94).

Remark 20: Line 790: *wer* should be *were*.

- ✓ The related sentence has been slightly modified, resulting in the disappearance of this typo.

Remark 21: Line 862: No *D* in Fig. S6.

- ✓ This mention originally appeared in the discussion, which has been considerably modified in the current version of the manuscript. Therefore, it no longer exists. The new paragraph related to the disruption of lipid homeostasis lies between lines 1183 and 1187.

Remark 22: Line 1300-1316: Fig.7 B and C become D and E in legend.

- ✓ The numbering was indeed not correct. In the present version, Figure 7 became Figure 8, and changes were made between the panels that compose it. We checked that the legend was consistent with the figure (lines 1725-57)

Remark 23: Figure 1: *c.854G>A p.(Gly282Asp)* should be *c.854G>A p.(Gly282Asp)*.

- ✓ The supplementary bracket was removed.

Remark 24: Only the first letter of the gene name of the model animals (*fly* and *rat*) are capitalized. All characters are capitals in Fig.3.

- ✓ The nomenclature of genes and proteins was modified in Fig. 3 to comply with the official rules for each organism: (i) for *Drosophila melanogaster*, gene in lower case and italics; (i) for *Ratus norvegicus*, gene italicized beginning with an uppercase letter followed by all lowercase letters, and protein symbols using all uppercase letters not italicized.

Remark 25: Possible multigenic origin could be commented regarding S11 as the similar finding was reported in PMID: 37600812 (if applicable) as the same authors are involved.

- ✓ This is indeed a possible mechanism that we evoked in the results section, where we report this specific case (lines 978-80). The reference suggested by the reviewer is especially relevant, because one of the cases presented involved a potential trigenic mechanism, with the contribution of a *PSMC5* splice site variant.

*Remark 26: What are N50Rfs*7, L52P, M130I?, V362M and D394G in Figure S1?*

- ✓ We thank the reviewer for noting these errors. These variants escaped our attention when we cleansed the data from the various figures in the manuscript. They had been included in first place, as they had been found in patients whose families eventually did not want to participate in the study. We corrected the figure, so that only the variants included appear.

Remark 27: This reviewer does not understand what gnomAD (1) and gnomAD (2) in Table 1 mean. please fill in the allele frequency of gnomAD as Table S1.

- ✓ As mentioned above, we deleted Table 1 to avoid redundancies with Figure 1 and Table S1. The allele frequencies from gnomAD are indicated in Table S1.

Remark 28: p.(Gly282Asp) is not shown in the 3D structural analysis figures.

- ✓ The variant was added to the figure.

Remark 29: In Fig.7 B, S12's father is a typo of S11's father.

- ✓ We corrected the labels in Fig. 8C (formerly Fig. 7B). Due to the change in numbering, "S11's father" is now "S14's father".

Remark 30: In Fig.7 C, color-code or reshape the variants so that you can see the score of each variant. The right panel of Figure S5 B and the line graph below C should be similarly shown as indicated above.

- ✓ We modified the color code in Fig. 8C-D (formerly Fig. 7B-C). We used a specific color for each patient and for S14's father who harbors the *PSMD11* variant p.(Arg90*).
- ✓ We also modified Fig. S6A-B (formerly S5A-B) so that the fold change of *PSMC5* levels can be detailed for each variant.

RESPONSE TO REFEREES

We would like to thank once again the referees for their thorough review and for recommending the manuscript for publication to the editor. Since Reviewer #1 was satisfied with all the changes made in the prior version, we detailed below only the remaining minor comments made by Reviewer #2.

Our point-by-point responses to the comments made are listed below.

Reviewer #2:

Page 6, line 247: Its. Knockdown>Its knockdown

- ✓ We thank the reviewer for spotting this typo, which has been removed from the manuscript as the sentence has been rewritten.

Page 27, line 807: (7/44 subjects)>(6/44 subjects) and (8/44)>(9/44) based on the new Figure 1A

- ✓ These figures had indeed not been updated as they should have been. This has now been done.

Figure 1B: Arg325Trp 8>9 and Pro320His 7>6 based on the new Figure 1A

- ✓ As with the previous comment, we have corrected this error too.

Page 27, line 817-818: the American College of Medical Genetics and Genomics (ACMG) guidelines>the American College of Medical Genetics and Genomics/the Association for Molecular Pathology (ACMG/AMP) guidelines

- ✓ We have followed this suggestion and indicated the full name for the ACMG/AMP.

Page 27, line 821: Fig. 1A>Fig. 2A

- ✓ We have corrected this error.

Page 32, line 952: (p.Arg325Trp)>p.(Arg325Trp)

- ✓ We moved the bracket to the right place.

Figure 5C: S23 (p.Pro320Arg)>S25 (p.Pro320Arg)

- ✓ We have modified the label in the figure.

Figure 6A: S7 (V7)>S7 (V6)

- ✓ We have modified this label too.

Figure 7B: gene name is hard to see. This reviewer suggests a bit enlarged version is recommended.

- ✓ We have redrawn the figure, in order to use larger font sizes.

Figure 7E legend: six NDD subjects, but this reviewer sees 7 red points in Figure 7E.

- ✓ We are grateful to the reviewer for noticing this discrepancy. We had indeed forgotten to indicate one patient in the legend. We have now listed six patients: S1, S7, S14, S15, S25, S36, and S37.